# Positioning of nucleosomes containing γ-H2AX precedes active DNA demethylation and transcription initiation

Stephanie Dobersch [1,2,3], Karla Rubio [1,2,4,5], Indrabahadur Singh[2,6], Stefan Günther[7,8], Johannes Graumann [9], Julio Cordero[10,11], Rafael Castillo-Negrete[1,2], Minh Bao Huynh[12], Aditi Mehta[2,13], Peter Braubach [14,15], Hector Cabrera-Fuentes[16,17,18,19], Jürgen Bernhagen [20,21], Cho-Ming Chao[22,23,24,25], Saverio Bellusci[22,23,24,25], Andreas Günther [24,25,26,27], Klaus T. Preissner[16,24], Sita Kugel[3], Gergana Dobreva [10,11], Malgorzata Wygrecka [24,25,28], Thomas Braun [8,24], Dulce Papy-Garcia[12] & Guillermo Barreto [1,2,4,5,24,25✉]

In addition to nucleosomes, chromatin contains non-histone chromatin-associated proteins, of which the high-mobility group proteins are the most abundant. Chromatin-mediated regulation of transcription involves DNA methylation and histone modifications. However, the order of events and the precise function of high-mobility group proteins during transcription initiation remain unclear. Here we show that high-mobility group AT-hook 2 protein (HMGA2) induces DNA nicks at the transcription start site, which are required by the histone chaperone FACT complex to incorporate nucleosomes containing the histone variant H2A.X. Further, phosphorylation of H2A.X at S139 (γ-H2AX) is required for repair-mediated DNA demethylation and transcription activation. The relevance of these findings is demonstrated within the context of TGFB1 signaling and idiopathic pulmonary fibrosis, suggesting therapies against this lethal disease. Our data support the concept that chromatin opening during transcriptional initiation involves intermediates with DNA breaks that subsequently require DNA repair mechanisms to ensure genome integrity.

A list of author affiliations appears at the end of the paper.

In the eukaryotic cell nucleus, chromatin is the physiological template of all DNA-dependent processes including transcription. The structural and functional units of chromatin are the nucleosomes, each one consisting of ~147 bp of genomic DNA wrapped around a core histone octamer, which in turn is built of two H2A–H2B dimers and one (H3–H4)$_2$ tetramer[1,2]. In addition to canonical histones (H1, H2A, H2B, H3 and H4), there are so called histone variants for all histones except for H4. Histone variants differ from the canonical histones in their amino acid sequence and have specific and fundamental functions that cannot be performed by canonical histones. The canonical histone H2A has a large number of variants, each with defined biochemical and functional properties[3,4]. Here we focus on the histone variant H2AFX (commonly known as H2AX, further referred to as H2A.X), which represents about 2–25% of the cellular H2A pool in mammals[5]. Phosphorylated H2A.X at serine 139 (H2A.XS139ph; commonly known as γ-H2AX, further referred to as pH2A.X) is used as a marker for DNA double-strand breaks[6]. However, accumulating evidence suggests additional functions of pH2A.X[7–10]. The histone chaperone FACT (facilitates chromatin transcription) is a heterodimeric complex, consisting of SUPT16 and SSRP1 (Spt16 and Pob3 in yeast) that is responsible for the deposition of H2A/H2B-dimers onto DNA[11,12]. The FACT complex mainly interacts with H2B mediating the deposition of H2A/H2B-dimers containing different H2A variants[13]. Thus, the deposition of H2A.X into chromatin seems to be mediated by the FACT complex[14].

In addition to nucleosomes, chromatin contains non-histone chromatin-associated proteins, of which the high-mobility group (HMG) proteins are the most abundant. Although HMG proteins do not possess intrinsic transcriptional activity, they are called architectural transcription factors because they modulate the transcription of their target genes by altering the chromatin structure at the promoter and/or enhancers[15]. Here we will focus on HMG AT-hook 2 protein (HMGA2), a member of the HMGA family that mediates transforming growth factor beta 1 (TGFB1, commonly known as TGFβ1) signaling[16]. We have previously shown that HMGA2-induced transcription requires phosphorylation of H2A.X at S139, which in turn is mediated by the protein kinase ataxia telangiectasia mutated (ATM)[10]. Furthermore, we demonstrated the biological relevance of this mechanism of transcriptional initiation within the context of TGFB1 signaling and epithelial-mesenchymal transition (EMT). Interestingly, TGFB signaling has been reported to induce active DNA demethylation with the involvement of thymidine DNA glycosylase (TDG)[17]. Active DNA demethylation also requires GADD45A (growth arrest and DNA damage protein 45 alpha) and TET1 (ten-eleven translocation methylcytosine dioxygenase 1), which sequentially oxidize 5-methylcytosine (5mC) to 5-carboxylcytosine (5caC)[18,19] and are cleared through DNA repair mechanisms. In line with these ideas, classical DNA-repair complexes have been linked to DNA demethylation and transcriptional activation[20,21]. On the other hand, DNA double-strand breaks induce ectopic transcription that is essential for repair, supporting a tight mechanistic correlation between transcription, DNA damage, and repair[22].

TGFB1 signaling and EMT are both playing a crucial role in idiopathic pulmonary fibrosis (IPF). IPF is the most common interstitial lung disease showing a prevalence of 20 new cases per 100,000 persons per year[23,24]. A central event in IPF is the abnormal proliferation and migration of fibroblasts in the alveolar compartment in response to lung injury. IPF patients die within 2 years after diagnosis mostly due to respiratory failure. Current treatments against IPF aim to ameliorate patient symptoms and to delay disease progression[25]. Unfortunately, therapies targeting the causes of or reverting IPF have not yet been developed. Here, we demonstrate that inhibition of the HMGA2-FACT-ATM-pH2A.X axis reduces fibrotic hallmarks in vitro using primary human lung fibroblast (hLF) and ex vivo using human precision-cut lung slices (hPCLS), both from control and IPF patients. Our study supports the development of therapeutic approaches against IPF using FACT inhibition.

## Results

**HMGA2 is required for pH2A.X deposition at transcription start sites.** We have previously reported that HMGA2-mediated transcription requires phosphorylation of the histone variant H2A.X at S139, which in turn is catalyzed by the protein kinase ATM[10]. To further dissect this mechanism of transcription initiation, we decided first to determine the effect of *Hmga2*-knockout (KO) on genome wide levels of pH2A.X. We performed next generation sequencing (NGS) after chromatin immunoprecipitation (ChIP-seq; Fig. 1 and Supplementary Fig. 1a) using pH2A.X-specific antibodies and chromatin isolated from mouse embryonic fibroblasts (MEF) from wild-type (WT = *Hmga2* + /+) and *Hmga2*-deficient (*Hmga2*-KO = *Hmga2*−/−)[26] embryos. The analysis of these ChIP-seq results using the UCSC Known Genes dataset[27] revealed that pH2A.X is specifically enriched at transcription start sites (TSS) of genes in an *Hmga2*-dependent manner (Fig. 1a), since *Hmga2*-KO significantly reduced pH2A.X levels. A zoom into the −750 to +750 base pair (bp) region relative to the TSS (Fig. 1b) revealed that pH2A.X levels significantly peaked ($\tilde{x}$ = 0.396; $n$ = 9522; $P < 2.2E$-16) at the TSS (−250 to +250 bp) in *Hmga2* + /+ MEF. Further, the genes were ranked based on pH2A.X levels at the TSS (Source Data file 04) and the results were visualized as heat maps (Fig. 1c). From the top 15% of the genes with high pH2A.X levels at TSS (further referred as top 15% candidates; $n$ = 9522), we selected *Gata6* (GATA binding protein 6), *Mtor* (mechanistic target of rapamycin kinase) and *Igf1* (insulin like growth factor 1) for further single gene analysis. Explanatory for these gene selection, we have previously reported *Gata6* as direct target gene of HMGA2[10,28], KEGG (Kyoto Encyclopedia of Genes and Genomes) pathway enrichment based analysis of the top 15% candidates showed significant enrichment of genes related to the mammalian target of rapamycin (mTOR) signaling pathway (Supplementary Fig. 1b), HMGA2 has been related to the insulin signaling pathway[29,30]. In addition, *Rptor* (regulatory associated protein of MTOR complex 1) is outside of the top 15% candidates (Fig. 1c) and was selected as negative control. Visualization of the selected genes using the UCSC genome browser confirmed the reduction of pH2A.X at specific regions close to TSS in *Hmga2*−/− MEF when compared to *Hmga2* + /+ MEF (Fig. 1d, top) except in the negative control *Rptor*. Similar results were obtained after ChIP-seq using H2A.X and H3 antibodies (Fig. 1d, bottom), the last one frequently used for monitoring nucleosome position. Promoter analysis of *Gata6*, *Mtor*, *Igf1* and *Rptor* by ChIP using pH2A.X-, H2A.X-, H3- and HMGA2-specific antibodies (Supplementary Fig. 1c, d) confirmed the ChIP-seq data. These findings suggest that the first nucleosome relative to the TSS of the top 15% candidates contains pH2A.X and *Hmga2* is required for correct positioning of this first nucleosome.

**Position of the first nucleosome containing pH2A.X correlates with RNA polymerase II and the basal transcription activity of genes.** Phosphorylation of specific amino acids in the C-terminal domain of the large subunit of the RNA polymerase II (Pol II) determines its interaction with specific factors, thereby regulating the transcription cycle consisting of initiation, elongation and termination[31]. To monitor transcription initiation, ChIP-seq was performed using antibodies specific for transcription initiating S5 phosphorylated Pol II (further referred to as pPol II) and

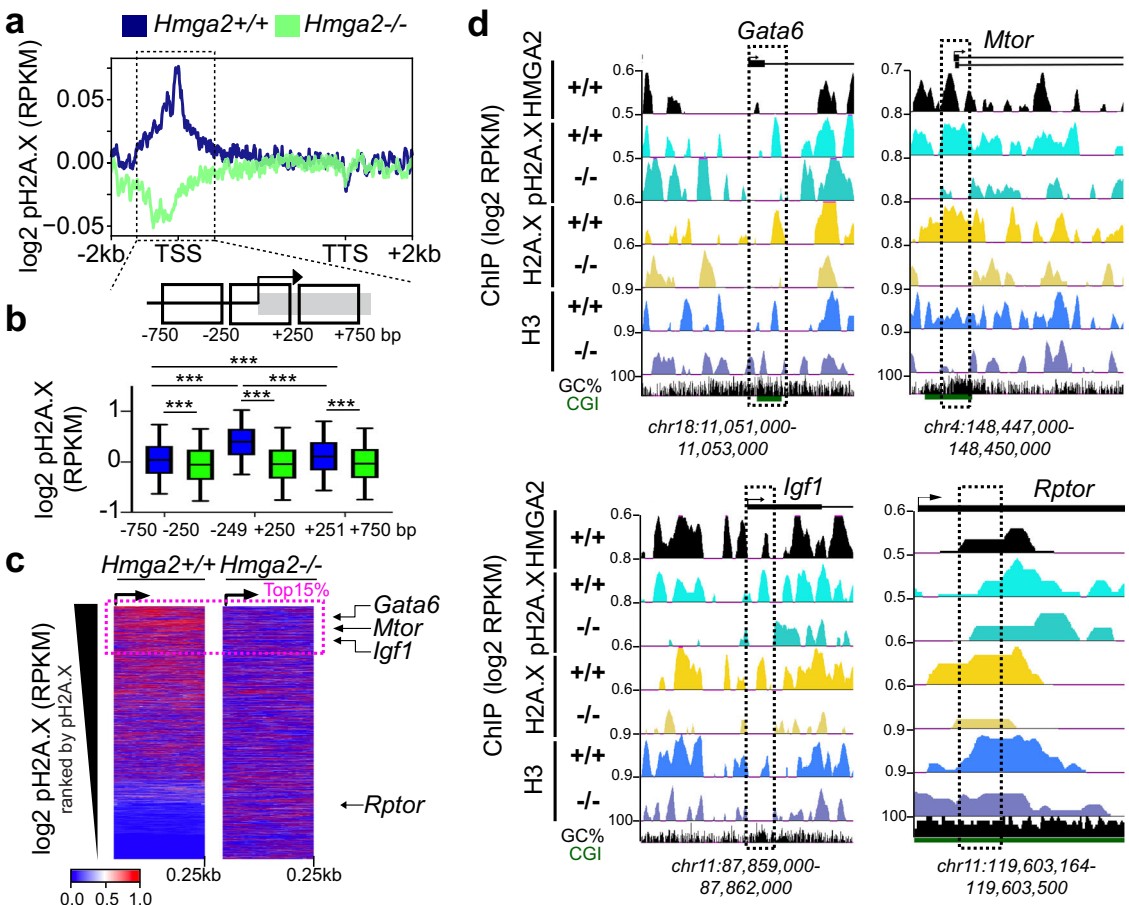

**Fig. 1 HMGA2 is required for pH2A.X deposition at TSS. a** Aggregate plot for pH2A.X enrichment within the gene body ±2 kb of UCSC Known Genes in *Hmga2*+/+ and *Hmga2*−/− MEF. ChIP-seq reads were normalized using reads per kilobase per million (RPKM) measure and are represented as log2 enrichment over their corresponding inputs. TSS, transcription start site; TTS, transcription termination site. Dotted square, ±750 bp region around the TSS. **b** Top, schematic representation of the genomic region highlighted in a. Bottom, box plot of pH2A.X enrichment in the genomic regions showed as squares at the top in *Hmga2*+/+ and *Hmga2*−/− MEF. RPKM of the pH2A.X ChIP-seq were binned within each of these genomic regions and represented as log2. Box plots indicate median (middle line), 25th, 75th percentile (box) and 5th and 95th percentile (whiskers); n = 9522 genes enriched with pH2A.X; asterisks P-values after two-tailed Wilcoxon–Mann–Whitney test, ***P ≤ 0.001. **c** Heat map for pH2A.X enrichment at the TSS + 0.25 kb of UCSC Known Genes in *Hmga2* + /+ and *Hmga2*−/− MEF. Genes were ranked by pH2A.X enrichment in *Hmga2* + /+ MEF. Doted square, the top 15% ranked genes, as well as *Gata6*, *Mtor*, *Igf1* and *Rptor* were selected for further analysis. **d** Visualization of selected HMGA2 target genes using UCSC Genome Browser showing HMGA2 (black), pH2A.X (turquoise), H2A.X (yellow) and H3 (blue) enrichment in *Hmga2*+/+ and −/− MEF. ChIP-seq reads were normalized using RPKM measure and are represented as log2 enrichment over their corresponding inputs. Images show the indicated gene loci with their genomic coordinates. Arrows, direction of the genes; black boxes, exons; dotted squares, regions selected for single gene analysis. See also Supplementary Fig. 1. Source data are provided as a Source Data files 01 and 04.

chromatin isolated from *Hmga2*+/+ and *Hmga2*−/− MEF (Fig. 2 and Supplementary Fig. 1a). Analysis of the ChIP-seq results using the UCSC Known Genes dataset revealed that pPol II was enriched at TSS in *Hmga2*-dependent manner (Fig. 2a). In addition, we observed that pPol II enrichment coincides with pH2A.X peaks at TSS (Fig. 2b) also in an *Hmga2*-dependent manner. Visualization of *Gata6*, *Mtor* and *Igf1* using the UCSC genome browser (Fig. 2c) and ChIP analysis of their promoters (Fig. 2d, left) confirmed the reduction of pPol II at specific regions close to TSS in *Hmga2*−/− MEF when compared to *Hmga2* + /+ MEF. Furthermore, the reduced pPol II levels after *Hmga2*-KO correlated with the reduced expression of the analyzed genes as shown by quantitative reverse-transcription PCR (qRT-PCR, Fig. 2d, right). These effects were not observed in the negative control *Rptor*. Moreover, *Hmga2* overexpression in *Hmga2*−/− MEF reverted the observed effects (Fig. 2d), thereby demonstrating the specificity of the changes caused by *Hmga2*-KO.

Further analysis of the ChIP-seq data by *k*-means clustering[32] revealed three clusters in the top 15% candidates (Fig. 3a). Cluster 1 (n = 3266) showed pPol II, pH2A.X, HMGA2 and H3 enrichment directly at the TSS (top), while clusters 2 (n = 3208) and 3 (n = 3075) showed enrichment of these proteins 125 bp and 250 bp 3′ of the TSS, respectively (middle and bottom). Remarkably, RNA sequencing (RNA-seq) based expression analysis in *Hmga2* + /+ and *Hmga2*−/− MEF (Fig. 3b) revealed that the genes in the three clusters have different basal transcription activities, whereby cluster 1 has the lowest (x̄ = 0.6201), cluster 2 the middle (x̄ = 0.6979) and cluster 3 the highest (x̄ = 1.56) basal transcription activity in *Hmga2* + /+ MEF. *Hmga2*-KO significantly reduced the basal transcription activity in all three clusters, to 0.325 (P = 4.72E-5) in cluster 1, 0.375 (P = 1E-4) in cluster 2 and 0.867 (P = 2.47E-3) in cluster 3. Our results demonstrate a correlation between the basal transcription activity and the position of pPol II, pH2A.X, HMGA2 and H3 relative to TSS. Interestingly, the observed *Hmga2*-

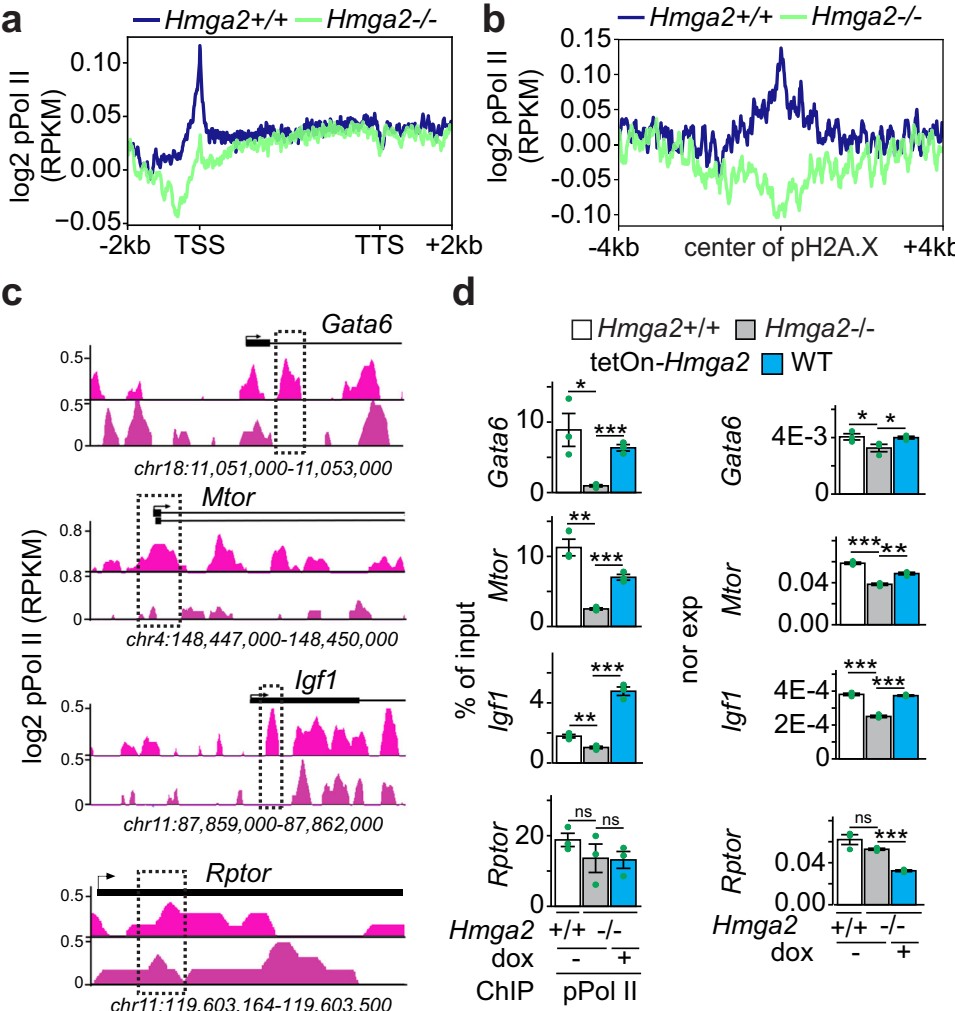

**Fig. 2 Position of transcription initiating S5 phosphorylated RNA polymerase II at the TSS is *Hmga2*-dependent. a, b** Aggregate plots for phosphorylated serine 5 RNA polymerase II (pPol II) enrichment within the gene body ±2 kb of UCSC Known Genes (**a**) and in a ± 4 kb region respective to pH2A.X peaks (**b**) in *Hmga2*+/+ and *Hmga2*−/− MEF. ChIP-seq reads were normalized using reads per kilobase per million (RPKM) measure and are represented as log2 enrichment over their corresponding inputs. TSS, transcription start site; TTS, transcription termination site. **c** Visualization of selected HMGA2 target genes using UCSC Genome Browser showing pPol II enrichment in *Hmga2* +/+ and −/− MEF. ChIP-seq reads were normalized using RPKM measure and are represented as log2 enrichment over their corresponding inputs. Images represent the indicated gene loci with their genomic coordinates. Arrows, direction of the genes; black boxes, exons; dotted squares, regions selected for single gene analysis. **d** Analysis of selected HMGA2 target genes. Left, ChIP of *Gata6, Mtor, Igf1* and *Rptor* after pPol II immunoprecipitation in *Hmga2* +/+ and *Hmga2*−/− MEF. Right, qRT-PCR-based, *Tuba1a*-normalized expression analysis under the same conditions. Bar plots presenting data as means; error bars, s.e.m (*n* = 3 biologically independent experiments); asterisks, *P*-values after two-tailed *t*-test, ***P* ≤ 0.001; **P* ≤ 0.01; **P* ≤ 0.05. See also Supplementary Figs. 1–2 . Source data are provided as a Source Data files 01 and 04.

dependent effects are specific for the top 15% candidates (*n* = 9522), which have relatively low pPol II levels and basal transcription activities (Supplementary Fig. 2a–c).

**HMGA2 is required for enrichment of the FACT complex at TSS.** To gain insights into the HMGA2, ATM, and pH2A.X transcriptional network[10], native chromatin preparations from *Hmga2* +/+ and *Hmga2*−/− MEF were digested with micrococcal nuclease (MNase) and subsequently fractionated by sucrose gradient ultracentrifugation (SGU) (Fig. 4a–e and Supplementary Fig. 3a–c). From the obtained fractions, the proteins were extracted and analyzed either by western blot (WB; Fig. 4a, d, top), by densitometry analysis of WB (Supplementary Fig. 3b), or by high-resolution mass spectrometry-based proteomic approach (Fig. 4b–c and Source Data file 04), while the DNA was

also isolated and analyzed either by gel electrophoresis (Fig. 4d, bottom), or by DNA sequencing (MNase-seq, Fig. 4e and Supplementary Fig. 4a, b). In *Hmga2* +/+ MEF (Fig. 4a, left and Supplementary Fig. 3b), WB of the obtained fractions showed that HMGA2 sedimented in fractions 4 to 9, whereas pPol II and histones mainly sedimented in fractions 1 to 4, where protein complexes of higher molecular weight (MW) are expected. Interestingly, the histone variant H2A.X and its post-translationally modified form pH2A.X showed a similar sedimentation pattern as the core histones. However, pH2A.X sedimentation in fraction 4 was more pronounced. In *Hmga2*−/− MEF (Fig. 4a, right and Supplementary Fig. 3b) the levels of pH2A. X were reduced and distributed in fractions 3 to 5. The reducing effect of *Hmga2*-KO on pH2A.X levels was confirmed by immunostaining in MEF (Supplementary Fig. 4c–d). The subsequent

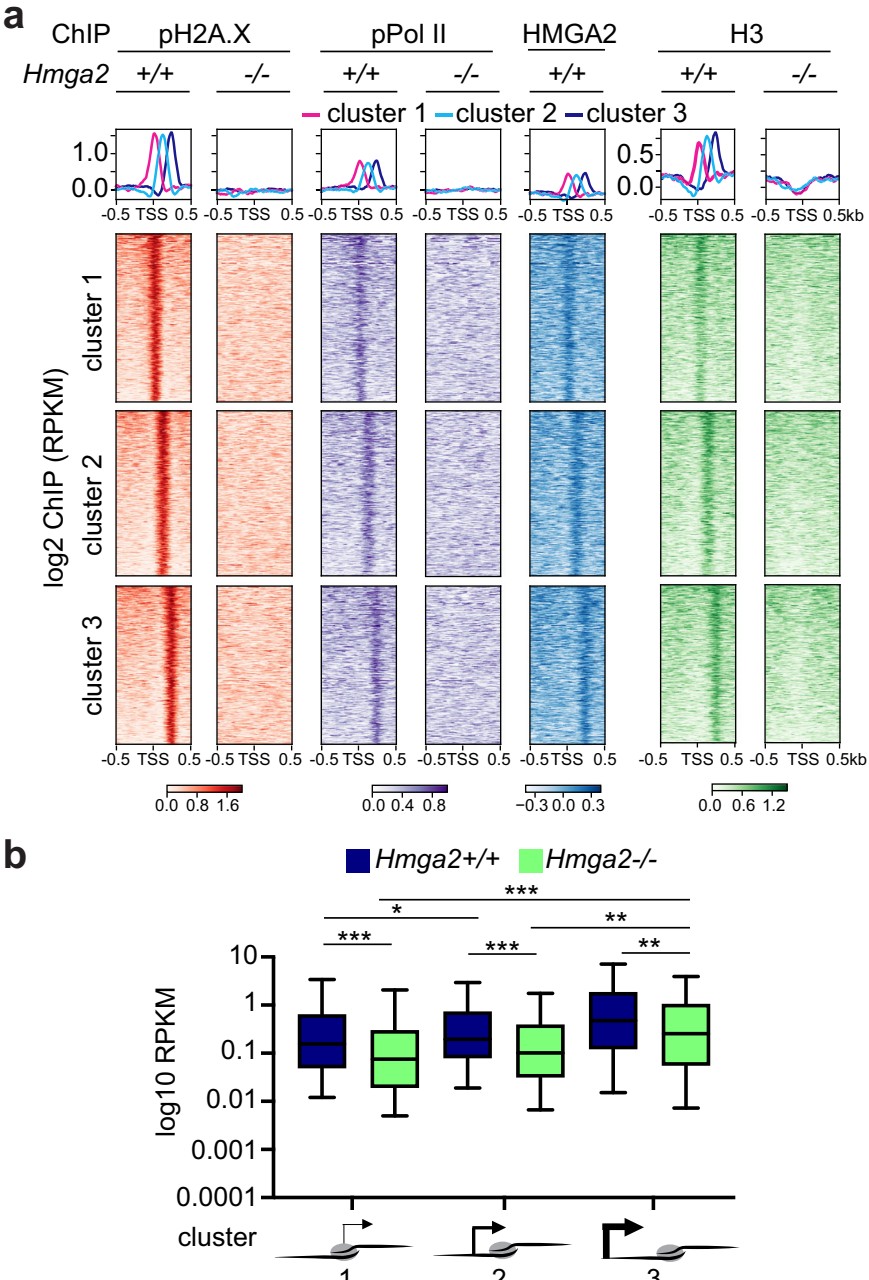

**Fig. 3 Position of first nucleosome containing pH2A.X correlates with RNA polymerase II and basal transcription activity in Hmga2-dependent manner. a** Aggregate plots (top) and heat maps (bottom) for pPol II, pH2A.X, HMGA2 and H3 enrichment at the TSS ± ≥ kb of the top 15% candidates in *Hmga2* +/+ and *Hmga2*−/− MEF. Three clusters were generated using *k*-means algorithm. Genes were sorted based on the enrichment of pH2A.X in *Hmga2* +/+ MEF. **b** Box plot representing the basal transcription activity (as log2 RPKM) of genes in *Hmga2* +/+ and *Hmga2*−/− MEF. The genes were sorted in three groups based on the position of the first pH2A.X-containing nucleosome 5′ to the TSS. Box plots indicate median (middle line), 25th, 75th percentile (box) and 5th and 95th percentile (whiskers); $n = 154$ genes in position cluster 1; $n = 132$ genes in position cluster 2; $n = 134$ genes in position cluster 3; asterisks, *P*-values after one-tailed Wilcoxon–Mann–Whitney test, ***$P \leq 0.001$; **$P \leq 0.01$; *$P \leq 0.05$. See also Supplementary Figs. 1 and 2. Source data are provided as a Source Data files 01 and 04.

analysis was focused on fractions 3 and 4, because fraction 4 contained all proteins monitored by WB in *Hmga2* +/+ MEF, whereas in fraction 3 levels of HMGA2 and pH2A.X were significantly reduced. We analyzed these two fractions by high-resolution mass spectrometry-based proteomic approach and identified proteins that were more than 1.65-fold significantly enriched in *Hmga2* +/+ MEF when compared to *Hmga2*−/− MEF (Fig. 4b; Source Data file 04; $n = 1215$ and $P < 0.05$ in fraction

3; $n = 1729$ and $P < 0.05$ in fraction 4). A closer look on the proteins enriched in fractions 3 and 4 of *Hmga2* +/+ MEF revealed the presence of both components of the FACT complex, SUPT16 and SSRP1 (Fig. 4b), as well as proteins related to transcription regulation and nucleotide excision repair (NER; Supplementary Fig. 3c). Interestingly, *Hmga2*-KO significantly reduced the levels of SUPT16 (from 0.909 to 0.298; $n = 3$; $P = 3.5E-3$) and SSRP1 (from 0.831 to 0.290; $n = 3$; $P = 3.9E-3$) in fraction 4 without

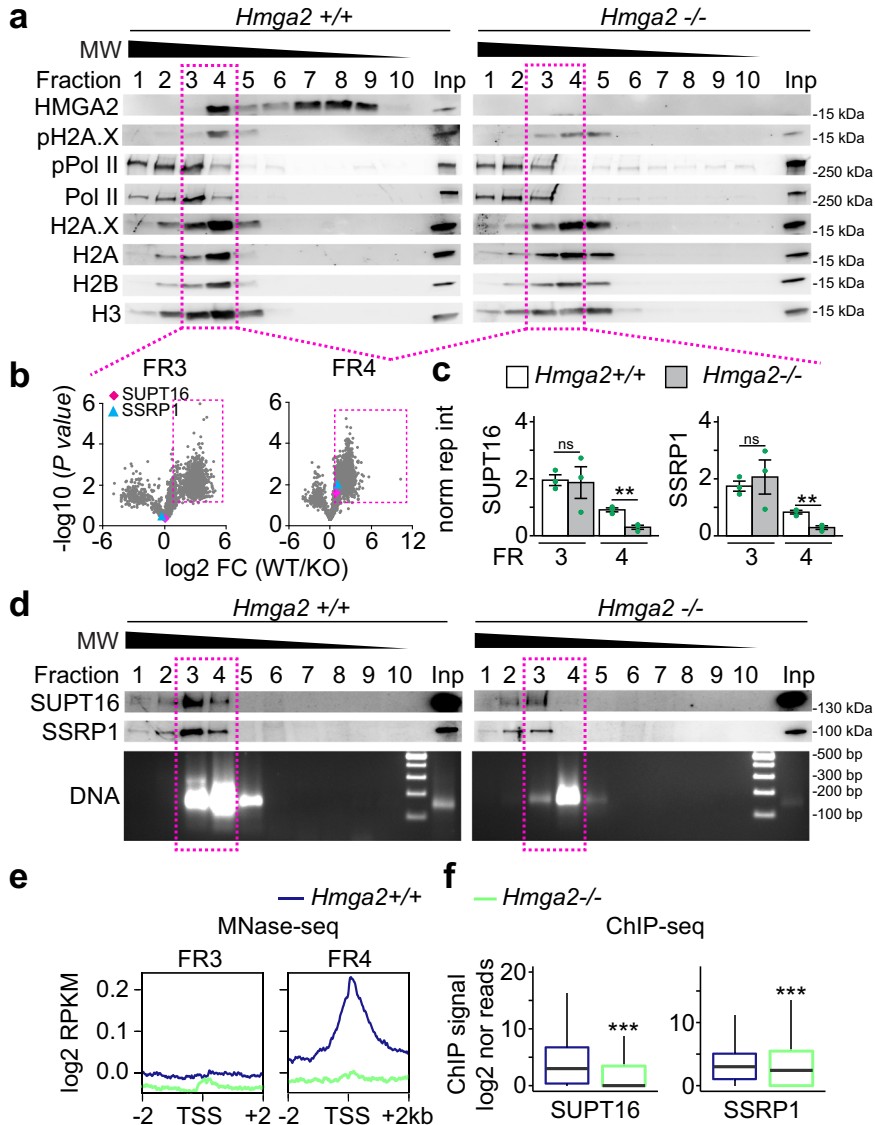

**Fig. 4 HMGA2 is required for enrichment of the FACT complex at TSS. a–e** Native chromatin from *Hmga2 +/+* and *−/−* MEF was digested with micrococcal nuclease (MNase) and fractionated by sucrose gradient ultracentrifugation (SGU). **a** The obtained fractions were analyzed by WB using the indicated antibodies. Representative images from three independent experiments. MW, molecular weight, kDa, kilo Dalton. Inp, input represents 0.5% of the material used for SGU. Square, fractions selected for further analysis. **b** Mass spectrometry analysis of proteins in fractions 3 and 4. Volcano plot representing the significance (−log10 *P*-values after one-tailed *t*-test) vs. intensity fold change between *Hmga2 +/+* and *−/−* MEF (log2 of means intensity ratios from three independent experiments). Square, proteins with log2 fold change >1.65. Diamond, SUPT16; triangle, SSRP1. **c** Bar plots showing normalized reporter intensity of SUPT16 (left) and SSRP1 (right) in fractions 3 and 4 of the SGU in a. Data are shown as means ± s.e.m. (*n* = 3 biologically independent experiments); asterisks, *P*-values after two-tailed *t*-Test, \*\**P* ≤ 0.01; ns, non-significant. **d** Top, WB analysis as in a using antibodies specific for components of the FACT complex. Bottom, DNA was isolated from the fractions obtained by SGU in A and analyzed by agarose gel electrophoresis. Representative images from three independent experiments. Square, fractions selected for MNase-seq. **e** MNase-seq of fractions 3 and 4 of the SGU in a. Aggregate plots representing the enrichment over input (as log2 RPKM) of genomic sequences relative to the TSS ± 2 kb. **f** Box plots of ChIP-seq-based SUPT16 (left) and SSRP1 (right) enrichment analysis within the TSS + 0.5 kb of the top 15% candidates in *Hmga2 +/+* and *−/−* MEF. Values are represented as log2 of mapped reads that were normalized to the total counts and the input was subtracted. Box plots indicate median (middle line), 25th, 75th percentile (box) and 5th and 95th percentile (whiskers); *n* = 9522 genes enriched with pH2A.X; asterisks, *P*-values after two-tailed Mann–Whitney test, \*\*\**P* ≤ 0.001. See also Supplementary Figs. 2 and 3. Source data are provided as a Source Data files 01 and 02.

significantly affecting their levels in fraction 3 (Fig. 4c). WB of SGU fractions using SUPT16- or SSRP1-specific antibodies (Fig. 4d, top) and the corresponding densitometry analysis (Supplementary Fig. 3b) show a reduction of both components of the FACT complex in fractions 3 and 4, being more pronounced in fraction 4. Further, we isolated and analyzed by electrophoresis the DNA from the SGU fractions and found DNA fragments with a length

profile of 100–300 bp that mainly sedimented in fractions 3 to 5 of both *Hmga2 +/+* and *Hmga2−/−* MEF. The length profile of DNA fragments supports the majority presence of mono- and dinucleosomes in the native chromatin preparations that were fractionated by SGU. Focusing again on fraction 3 and 4, MNase-seq was performed and reads between 100 to 200 bp were selected for downstream analysis (Fig. 4e and Supplementary Fig. 4a, b). Using

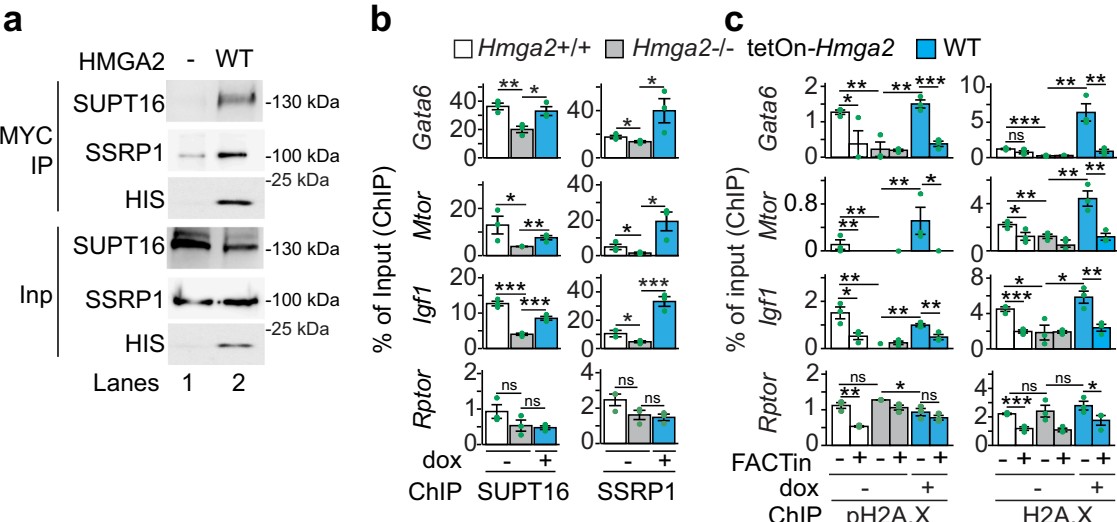

**Fig. 5 HMGA2 -FACT interaction is required for pH2A.X deposition. a** Western blot using the indicated antibodies after co-immunoprecipitation (Co-IP) assay using nuclear protein extracts from *Hmga2−/−* MEF that were non-transfected (−) or stably transfected with *Hmga2-myc-his* (WT) and magnetic beads coated with MYC-specific antibodies. Representative images from three independent experiments. Input, 5% of IP starting material. **b**, **c** ChIP-based promoter analysis of selected HMGA2 target genes using the indicated antibodies and chromatin from *Hmga2 + /+*, *Hmga2−/−* MEF, as well as *Hmga2−/−* MEF that were stably transfected with a tetracycline-inducible expression construct (tetOn) for WT *Hmga2-myc-his*. MEF were treated with doxycycline and FACT inhibitor (FACTin; CBLC000 trifluoroacetate) as indicated. In all bar plots, data are shown as means ± s.e.m. (*n* = 3 biologically independent experiments); asterisks, *P*-values after one-tailed *t*-Test, ***$P \leq 0.001$; **$P \leq 0.01$; *$P \leq 0.05$; ns, non-significant. See also Supplementary Fig. 5. Source data are provided as a Source Data files 01 and 02.

the UCSC Known Genes as reference dataset, we found that in fraction 4 the sequencing reads were enriched with TSS in an *Hmga2*-dependent manner (Fig. 4e), since this enrichment was abolished after *Hmga2*-KO. In fraction 3, we did not detect TSS enrichment of the sequencing reads. Our results indicate that the native chromatin in fraction 4 contains mono- and di-nucleosomes (Supplementary Fig. 4b), which are enriched with TSS, HMGA2, pH2A.X and pPol II in WT MEF. To link the results obtained by MNase-seq and mass spectrometry after fractionation by SGU, we performed ChIP-seq using SUPT16- and SSRP1-specific antibodies and chromatin from *Hmga2 + /+* and *Hmga2−/−* MEF (Fig. 4f and Supplementary Fig. 4e). Confirming the results in fraction 4, an accumulation of both components of the FACT complex was detected at TSS in *Hmga2 + /+* MEF, whereas *Hmga2*-KO reduced the levels of SUPT16 and SSRP1 at TSS. In summary, these results demonstrate that HMGA2 is required for recruitment of the FACT complex to TSS.

**HMGA2-FACT interaction is required for pH2A.X deposition**. The results in Fig. 4a–f suggest an interaction between HMGA2 and the FACT complex. In addition, we found in our previously published mass spectrometry based HMGA2 interactome[10] that HMGA2 precipitated SUPT16 and SSRP1 (Supplementary Fig. 4f). Indeed, the interaction of HMGA2 with both components of the FACT complex was confirmed by co-immunoprecipitation (Co-IP) assay using nuclear protein extracts from MEF after overexpression of HMGA2 tagged C-terminally with MYC and HIS (HMGA2-MYC-HIS; Fig. 5a). To further characterize the HMGA2-FACT interaction, nuclear extracts from *Hmga2 + /+* and *Hmga2−/−* MEF were fractionated into chromatin-bound and nucleoplasm fractions (Supplementary Fig. 5a). WB of these fractions showed that HMGA2 was exclusively bound to chromatin, whereas SUPT16 and SSRP1 were present in both sub-nuclear fractions. However, higher levels of both FACT components were detected in the chromatin-bound fraction of *Hmga2 + /+* MEF as compared to the

nucleoplasm. Interestingly, *Hmga2*-KO reverted the distribution of SUPT16 and SSRP1 between these two sub-nuclear fractions, thereby supporting that *Hmga2* is required for tethering the FACT complex to chromatin. These results were confirmed by ChIP of *Gata6*, *Mtor*, *Igf1* and *Rptor* (Fig. 5b) using SUPT16- and SSRP1-specific antibodies and chromatin isolated from *Hmga2 + /+* and *Hmga2−/−* MEF that were stably transfected with a tetracycline-inducible expression construct (tetOn) either empty (−; negative control) or containing the cDNA of WT HMGA2-MYC-HIS. Doxycycline treatment of these stably transfected MEF induced the expression of WT HMGA2-MYC-HIS (Supplementary Fig. 5b). *Hmga2*-KO reduced the levels of SUPT16 and SSRP1 in the promoters of the selected HMGA2 target genes (Fig. 5b), while doxycycline-inducible expression of WT HMGA2-MYC-HIS in *Hmga2−/−* MEF reconstituted the levels of both FACT components, thereby demonstrating the specificity of the effects caused by *Hmga2*-KO. These effects were not observed in the negative control *Rptor*.

To demonstrate the causal involvement of the FACT complex in context of HMGA2-mediated chromatin rearrangements, we analyzed the levels of pH2A.X and H2A.X at the *Gata6*, *Mtor*, *Igf1* and *Rptor* promoters by ChIP using chromatin from *Hmga2 + /+* MEF and stably transfected *Hmga2−/−* MEF that were treated with DMSO (control) or a FACT inhibitor (FACTin; CBLC000 trifluoroacetate)[33] and doxycycline as indicated (Fig. 5c). FACTin treatment induces negative supercoiling and the formation of left-handed Z-DNA, which is recognized by the FACT subunit SSRP1[34]. Increasing doses of FACTin resulted in chromatin trapping of SUPT16 and SSRP1 (Supplementary Fig. 5c, d)[35]. In addition, FACT inhibition significantly reduced pH2A.X and H2A.X levels specifically at the promoters of the selected HMGA2 target genes, confirming that the FACT complex is required for proper H2A.X deposition (Fig. 5c, left). Further, *Hmga2*-KO also reduced pH2A.X and H2A.X levels at the same promoters (middle), while doxycycline-inducible expression of WT HMGA2-MYC-HIS in *Hmga2−/−* MEF reconstituted the pH2A.X and H2A.X levels (right), thereby confirming the results

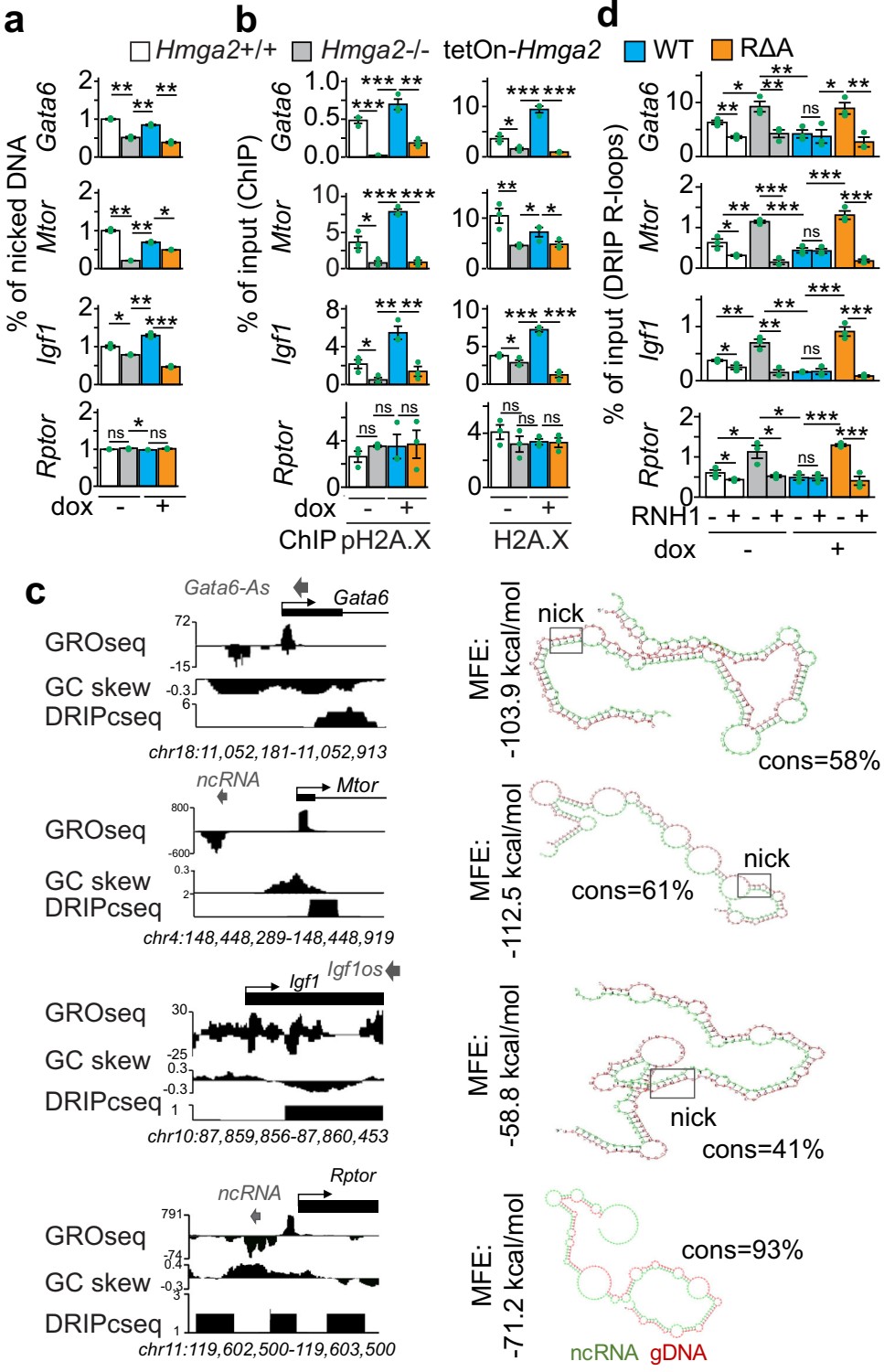

presented in Figs. 1 and 5b. Remarkably, FACT inhibition counteracted the reconstituting effect mediated by doxycycline-inducible expression of WT HMGA2-MYC-HIS, showing the causal involvement of the FACT complex in the function of HMGA2. In summary, these results demonstrate that the FACT complex is required for HMGA2 function and consequently also for proper pH2A.X levels at the promoters of the HMGA2 target genes.

After confirming the loss of lyase activity in RΔA HMGA2-MYC-HIS (Supplementary Fig. 6c, d), single-strand DNA breaks (DNA nicks) at the *Gata6*, *Mtor*, *Igf1* and *Rptor* promoters were monitored using genomic DNA from *Hmga2+/+* MEF and stably transfected *Hmga2−/−* MEF that were non-treated (−) or treated with doxycycline (Fig. 6a). We detected DNA nicks at the analyzed promoters in *Hmga2+/+* MEF, whose levels were reduced upon *Hmga2*-KO specifically at the promoter of the

**Fig. 6 HMGA2-lyase activity is required for pH2A.X deposition and solving of R-loops. a** Analysis of single-strand DNA breaks at promoters of selected HMGA2 target genes using genomic DNA from $Hmga2+/+$, $Hmga2-/-$ MEF, as well as $Hmga2-/-$ MEF that were stably transfected with a tetracycline-inducible expression construct (tetOn) either for WT $Hmga2-myc-his$ or the lyase-deficient mutant RΔA $Hmga2-myc-his$. MEF were treated with doxycycline as indicated. **b** ChIP-based promoter analysis of selected HMGA2 target genes using the indicated antibodies and chromatin from MEF as in **a**. **c** Left, Genome-browser visualization of selected HMGA2 target genes showed nascent RNA (GRO-seq) in WT MEF[68], GC-skew and RNA-seq on plus-strand after DNA-RNA hybrid immunoprecipitation (DRIPc-seq) in NIH/3T3 mouse fibroblasts[40]. Images represent mapped sequence tag densities relative to the indicated loci. Genomic coordinates are shown at the bottom. Arrow heads, non-coding RNAs in antisense orientation; Arrows, direction of the genes; black boxes, exons. Right, in silico analysis revealed complementary sequences between the identified antisense ncRNA (green) and genomic sequences at the TSS of the corresponding mRNAs (red) with relatively favorable minimum free energy (MFE) and high percentage of complementarity (cons), supporting the formation of DNA-RNA hybrids containing a nucleotide sequence that favors DNA nicks (squares). **d** Analysis of selected HMGA2 target genes by DNA–RNA immunoprecipitation (DRIP) using the antibody S9.6[41] and nucleic acids isolated from MEF treated as in **a**. Prior IP, nucleic acids were digested with RNase H1 (RNH1) as indicated. In all bar plots, data are shown as means ± s.e.m. ($n=3$ biologically independent experiments); asterisks, $P$-values after two-tailed $t$-Test, ***$P \le 0.001$; **$P \le 0.01$; *$P \le 0.05$; ns, non-significant. See also Supplementary Figs. 5–7. Source data are provided as a Source Data file 01.

selected HMGA2 target genes. Interestingly, inducible expression of WT HMGA2 in $Hmga2-/-$ MEF reconstituted the levels of DNA nicks, whereas RΔA HMGA2 did not rescue the effect induced by $Hmga2$-KO, thereby confirming that HMGA2 lyase activity is required for the DNA nicks detected at the promoters of HMGA2 target genes. Further, we decided to demonstrate the requirement of the lyase activity for the function of HMGA2. Thus, the levels of pH2A.X and H2A.X at the $Gata6$, $Mtor$, $Igf1$ and $Rptor$ promoters were analyzed by ChIP using chromatin from $Hmga2+/+$ and stably transfected $Hmga2-/-$ MEF (Fig. 6b). Confirming the results in Figs. 1 and 5c, $Hmga2$-KO reduced pH2A.X and H2A.X levels specifically at the promoter of the selected HMGA2 target genes. In addition, doxycycline-inducible expression of WT HMGA2 in $Hmga2-/-$ cells reconstituted the levels of pH2A.X and H2A.X, demonstrating the specificity of the effect observed after $Hmga2$-KO. However, doxycycline-inducible expression of RΔA HMGA2 in $Hmga2-/-$ MEF did not rescue the effect induced by $Hmga2$-KO. Although the mutations inducing the loss of the lyase activity did not affect the HMGA2-FACT complex interaction (Supplementary Fig. 6e), these results show that the lyase activity is required for proper pH2A.X and H2A.X levels at the promoter of the selected HMGA2 target genes.

Interestingly, crossing the NONCODE database with our top 15% candidates revealed that 79% of the candidates have annotated noncoding RNAs (ncRNAs) in close proximity ($n = 7535$), including $Gata6$, $Mtor$, $Igf1$ and $Rptor$ (Fig. 6c, left top). Mapping the identified ncRNAs to the murine genome allowed us to identify 2,106 unique ncRNAs (7.4%) that mapped to loci close to promoters controlling the expression of adjacent mRNAs (Supplementary Fig, 7b). From these promoter related ncRNAs[36,37] more than half (1401; 67%) were in the antisense strand (as) in divergent (div; 621 ncRNAs) or convergent (con; 780 ncRNAs) orientation[36,37] relative to the corresponding promoter and mRNA (Supplementary Fig. 7c, d). Interestingly, $Hmga2$-KO significantly reduced the median expression levels of these antisense divergent ncRNAs from 0.085 to 0.067 ($P = 0.039$; Supplementary Fig. 7e), without significantly affecting the levels of antisense convergent and sense ncRNAs. In silico analysis allowed us to detect putative binding sites of the identified ncRNAs at the TSS of the corresponding mRNAs (Fig. 6c, right) with favorable minimum free energy (MFE $< -55$ kcal/mol) and relatively high consensus (cons $> 41\%$;), supporting the formation of DNA-RNA hybrids containing a nucleotide sequence that favors DNA nicks[38]. In the same genomic regions, we also identified strand asymmetry in the distribution of cytosines and guanines, so called GC skews (Fig. 6c, left middle; Supplementary Fig. 7f, g), that are predisposed to form R-loops, which are three-stranded nucleic acid structures consisting of a DNA-RNA hybrid

and the associated non-template single-stranded DNA[39]. Supporting this hypothesis, published genome-wide sequencing experiments after DNA–RNA immunoprecipitation (DRIP-seq) in NIH/3T3 mouse fibroblasts[40] confirmed the formation of DNA-RNA hybrids in the top 15% candidate genes ($n = 9522$; Supplementary Fig. 7f, g), including $Gata6$, $Mtor$, $Igf1$ and $Rptor$ (Fig. 6c, left bottom). This correlated with high amounts of GC skews at their TSS with at least 38.5% of the TSS and downstream region having a GC skew higher than 0.05 ($n = 3669$). All these observations prompted us to investigate the role of HMGA2 during R-loop formation at the TSS. Thus, we analyzed by DRIP assays the levels of R-loops at the $Gata6$, $Mtor$, $Igf1$ and $Rptor$ promoters using the antibody S9.6[41] and nucleic acids isolated from $Hmga2+/+$ and stably transfected $Hmga2-/-$ MEF (Fig. 6d). $Hmga2$-KO increased R-loops levels at the promoters analyzed, whereas doxycycline-inducible expression of WT HMGA2 in $Hmga2-/-$ MEF reduced R-loop levels back to similar levels as in $Hmga2+/+$ MEF. Interestingly, doxycycline-inducible expression of RΔA HMGA2 in $Hmga2-/-$ MEF did not rescue the effect induced by $Hmga2$-KO. In parallel, treatment of the samples before IP with RNase H1 (RNH1), which degrades RNA in DNA-RNA hybrids, reduced the levels of R-loops in all tested conditions, demonstrating the specificity of the antibody S9.6[41]. In summary, these results demonstrate that HMGA2 and its lyase activity are required to solve R-loops at the analyzed promoters, including the negative control $Rptor$. The fact that we detected R-loops in $Rptor$ (Fig. 6d) without significant changes in the levels of DNA nicks (Fig. 6a) or pH2A.X (Fig. 6b) suggest a different regulatory mechanism for $Rptor$ when compared to the selected $Hmga2$ target genes.

**HMGA2-FACT-ATM-pH2A.X axis is required to solve R-loops and induce DNA demethylation.** The inducible expression of RΔA HMGA2 in $Hmga2-/-$ MEF did not decrease R-loops levels at TSS that were increased after $Hmga2$-KO (Fig. 6d), supporting that the lyase activity of HMGA2 is required to solve R-loops. To further investigate these results, the levels of double-stranded DNA (dsDNA) at the $Gata6$, $Mtor$ and $Igf1$ promoters were analyzed by DNA immunoprecipitation (DIP) assays (Fig. 7a, left). Inversely correlating with the effects on R-loops, $Hmga2$-KO reduced dsDNA levels at the promoters analyzed. Further, doxycycline-inducible expression of WT HMGA2 in $Hmga2-/-$ MEF reconstituted dsDNA levels, whereas RΔA HMGA2 failed to rescue the effect induced by $Hmga2$-KO. These results further support the requirement of HMGA2 and its lyase activity for solving R-loops. Since DNA methylation alters chromatin structure and is associated with R-loop formation[18,42], we also analyzed the levels of 5-methylcytosine (5mC) at the $Gata6$, $Mtor$ and $Igf1$ promoters by DIP assays using 5mC-

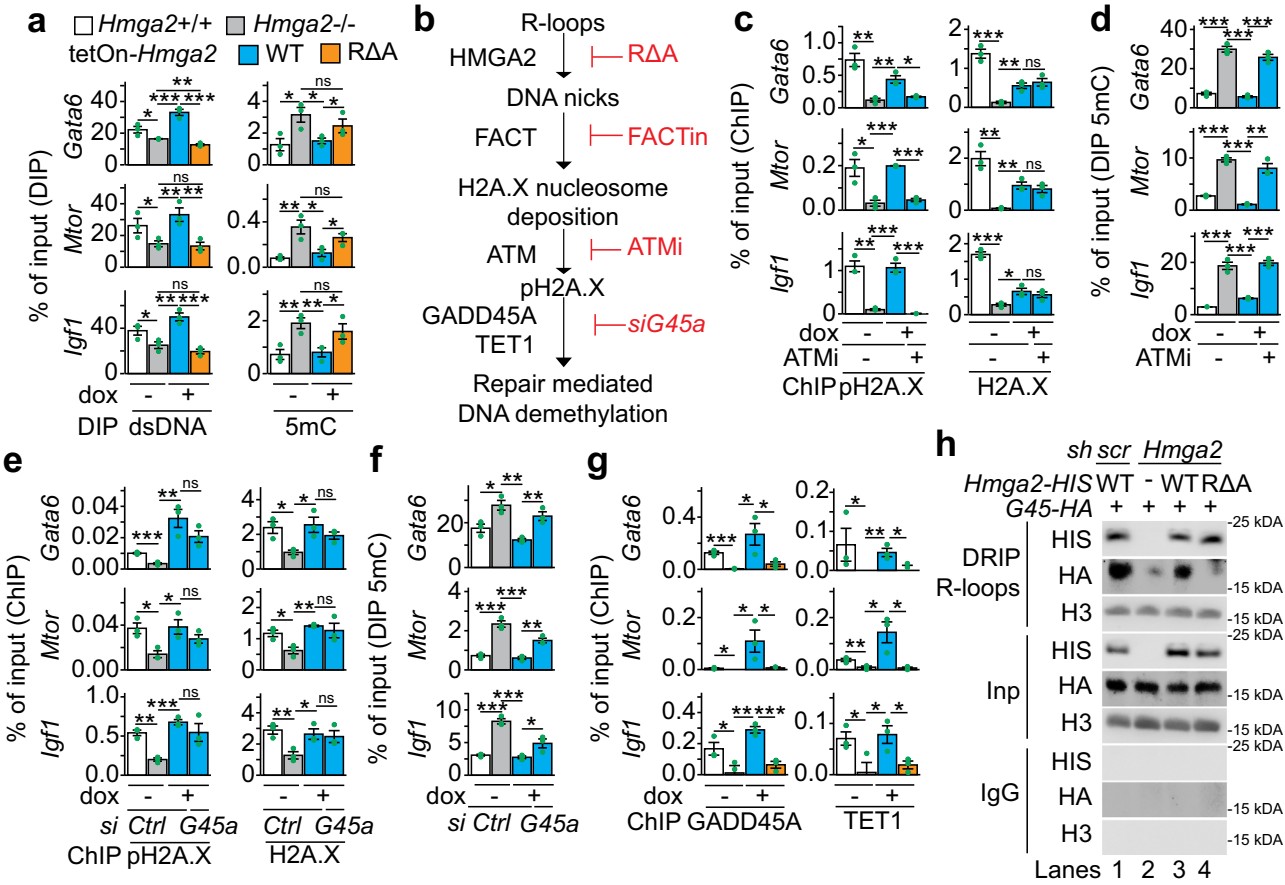

**Fig. 7 HMGA2-FACT-ATM-pH2A.X axis is required to solve R-loops and induce DNA demethylation. a** DNA immunoprecipitation (DIP) based promoter analysis of selected HMGA2 target genes using antibodies specific for double-stranded DNA (dsDNA) or 5-methylcytosine (5mC) and genomic DNA from *Hmag2+/+*, *Hmga2−/−* MEF, as well as *Hmga2−/−* MEF that were stably transfected with a tetracycline-inducible expression construct (tetOn) for either WT *Hmga2-myc-his* or the lyase-deficient mutant RΔA *Hmga2-myc-his*. MEF were treated with doxycycline as indicated. **b** Schematic representation of the sequential order of events during transcription activation mediated by the HMGA2-FACT-ATM-pH2A.X axis. **c** ChIP-based promoter analysis of selected HMGA2 target genes using the indicated antibodies and chromatin from MEF treated as in a. In addition, MEF were treated with ATM inhibitor (ATMi; KU-55933) as indicated. **d** DIP-based promoter analysis as in a, using 5mC-specific antibodies. In addition, MEF were treated with ATMi as indicated. **e** ChIP-based promoter analysis of selected HMGA2 target genes using the indicated antibodies and chromatin from MEF treated as in a. In addition, MEF were transfected with control (Ctrl) or *Gadd45a*-specific small interfering RNA (siRNA) as indicated. **f** DIP-based promoter analysis as in a, using 5mC-specific antibodies. In addition, MEF were transfected with Ctrl or *Gadd45a*-specific siRNA as indicated. **g** ChIP-based promoter analysis of selected HMGA2 target genes using the indicated antibodies and chromatin from MEF treated as in a. **h** WB analysis using antibodies specific for HIS-tag, HA-tag and H3 after DRIP using the antibody S9.6[41] and chromatin isolated from MLE-12 cells that were stably transfected either with a control (scramble, *scr*) or an *Hmga2*-specific short hairpin DNA (*sh*) construct and transiently transfected with WT *Hmga2-myc-his* or the lyase-deficient mutant RΔA *Hmga2-myc-his* and *Gadd45*-HA as indicated. Representative image from two independent experiments. Input (Inp), 5% of IP starting material; immunoglobulin G (IgG), negative control. In all bar plots, data are shown as means ± s.e.m. (*n* = 3 biologically independent experiments); asterisks, *P*-values after one-tailed *t*-test, ***P ≤ 0.001; **P ≤ 0.01; *P ≤ 0.05; ns, non-significant. See also Supplementary Fig. 8. Source data are provided as a Source Data files 01 and 02.

specific antibodies[43] (Fig. 7a, right). Correlating with the effects on R-loops, *Hmga2*-KO increased 5mC levels, which in turn were reduced by inducible expression of WT HMGA2 in *Hmga2−/−* cells but not by RΔA HMGA2. These results showed that HMGA2 and its lyase activity are required for proper 5mC levels at the analyzed promoters. In addition, the results using FACTin (Fig. 5c) showed that the FACT complex is required for HMGA2 function and consequently for proper pH2A.X levels at TSS.

To demonstrate the sequential order of events of the molecular mechanism proposed here (Fig. 7b), additional experiments were performed (Fig. 7 and Supplementary Fig. 8). We first analyzed the *Gata6*, *Mtor* and *Igf1* promoters by DRIP and DIP using nucleic acids isolated from *Hmga2+/+* MEF that were non-treated (-) or treated with FACTin as indicated (Supplementary Fig. 8a, b). FACTin treatment in *Hmga2+/+* MEF increased R-loop and 5mC levels, whereas dsDNA levels were reduced,

thereby supporting that the FACT complex is required to solve R-loops and for proper levels of 5mC at the analyzed promoters, similarly as the HMGA2 lyase activity (Figs. 6d and 7a). Previously, we have shown that ATM loss-of-function (LOF) blocks TGFB1-induced and HMGA2-mediated transcription activation[10]. To confirm the causal involvement of ATM in the mechanism of transcription regulation proposed here (Fig. 7b), the levels of pH2A.X and H2A.X at the *Gata6*, *Mtor* and *Igf1* promoters were analyzed by ChIP using chromatin from *Hmga2+/+* MEF and stably transfected *Hmga2−/−* MEF that were treated with DMSO (control) or an ATM inhibitor (ATMi; KU-55933) and doxycycline as indicated (Fig. 7c). Interestingly, ATMi treatment counteracted the rescue effect on pH2A.X levels that was mediated by inducible expression of WT HMGA2 in *Hmga2−/−* MEF, without significantly affecting H2A.X levels, thereby supporting that ATM is required for the post-

translational modification of H2A.X rather than for the deposition of H2A.X into the analyzed promoters. In addition, we monitored 5mC levels by DIP at the *Gata6*, *Mtor* and *Igf1* promoters (Fig. 7d) and found that ATM-LOF also counteracted the rescue effect on 5mC levels mediated by inducible expression of WT HMGA2 in *Hmga2*−/− MEF, thereby supporting that phosphorylation of H2A.X at S139 is required for proper 5mC levels. The results obtained after ATM-LOF (Fig. 7c, d) support that ATM acts downstream of HMGA2 and the FACT complex.

*Gadd45a* has been reported to promote transcriptional activation by repair-mediated DNA demethylation[19]. Thus, we investigated the potential involvement of *Gadd45a* in the order of events proposed here (Fig. 7b). To this purpose, we analyzed the effect of *Gadd45a*-specific LOF using small interfering RNA (siRNA; siG45a; Supplementary Fig. 8c) on pH2A.X, H2A.X and 5mC levels at the *Gata6*, *Mtor* and *Igf1* promoters in *Hmga2*+/+ MEF and stably transfected *Hmga2*−/− MEF (Fig. 7e, f). While siG45a transfection counteracted the rescue effect on 5mC levels mediated by inducible expression of WT HMGA2 in *Hmga2*−/− MEF (Fig. 7f), it did not significantly affect pH2A.X and H2A.X levels (Fig. 7e), confirming that GADD45A is required for proper 5mC levels but not for pH2A.X and H2A.X levels. Further, we found that GADD45A gain-of-function (GOF) after transfection of a human *GADD45A* expression construct into mouse lung epithelial (MLE-12) cells reduced 5mC levels in HMGA2-dependent manner (Supplementary Fig. 8d, e). Our results (Fig. 7e, f and Supplementary Fig. 8d, e) indicate that GADD45A acts downstream of the HMGA2-FACT-ATM-pH2A.X axis (Fig. 7b). Confirming this interpretation, ChIP-seq using GADD45A-specific antibodies and chromatin isolated from *Hmga2*+/+ and *Hmga2*−/− MEF (Supplementary Fig. 8f) revealed that GADD45A and pH2A.X are enriched at similar regions respective to TSS of the top 15% candidates. Moreover, ChIP analysis of the *Gata6*, *Mtor* and *Igf1* promoters using GADD45A- or TET1-specific antibodies and chromatin from *Hmga2*+/+ and stably transfected *Hmga2*−/− MEF that were treated with DMSO (control) or doxycycline (Fig. 7g) showed that *Hmga2*-KO abrogated GADD45A and TET1 binding to the analyzed promoters. Strikingly, inducible expression of WT HMGA2 reconstituted GADD45A and TET1 binding to the analyzed promoters, whereas RΔA HMGA2 did not rescue the effect induced by *Hmga2*-KO.

We have shown that genetic ablation of *Hmga2* increased R-loop levels (Fig. 6d) and reduced GADD45A binding (Fig. 7g) at the *Gata6*, *Mtor* and *Igf1* promoters. Interestingly, Arab and colleagues recently reported that GADD45A preferentially binds DNA-RNA hybrids and R-loops rather than single-stranded (ss) or double-stranded (ds) DNA or RNA[18]. To elucidate these at first glance contradictory results, we performed DRIP using the antibody S9.6[41] and chromatin from MLE-12 cells that were stably transfected with a scrambled (scr) or a *Hmga2*-specific short hairpin RNA construct (sh*Hmga2*) and non-treated (-) or treated with doxycycline to induce transient expression of WT or RΔA HMGA2 (Fig. 7h). WB analysis of the precipitated material revealed that both WT and RΔA HMGA2 bind to R-loops. Further, exogenous GADD45A also binds to R-loops, confirming the results by Arab and colleagues[18]. However, GADD45A binding to R-loops increased after inducible expression of WT HMGA2, but not after RΔA HMGA2, suggesting that DNA nicks in the R-loops increase the affinity of GADD45A to the R-loops. The results by WB after DRIP (Fig. 7h) correlate with the ChIP analysis of the *Gata6*, *Mtor* and *Igf1* promoters (Fig. 7g) and were confirmed by DRIP and sequential ChIP (DRIP-ChIP; Supplementary Fig. 8g). Taking together, our results demonstrate that the HMGA2-FACT-ATM-pH2A.X axis acts upstream of GADD45A and facilitates its binding to R-loops at specific promoters by nicking the DNA moiety of the DNA-RNA hybrid, thereby inducing DNA repair-mediated promoter demethylation.

**HMGA2-FACT-ATM-pH2A.X axis mediates TGFB1 induced transcription activation**. We have previously shown that HMGA2 mediates TGFB1 induced transcription[10]. Thus, we decided to evaluate the mechanism of transcription activation proposed here (Fig. 7b) within the context of TGFB1 signaling. We performed RNA-seq in *Hmga2*+/+ and *Hmga2*−/− MEF that were non-treated or treated with TGFB1 and visualized the results of those genes that were induced by TGFB1 treatment as heat maps after *k*-means clustering (Fig. 8a and Supplementary Fig. 9a). Four clusters were identified, of which clusters 2 ($n = 1471$) and 4 ($n = 1974$) contained genes that were TGFB1 inducible in an *Hmga2*-independent manner. Cluster 3 ($n = 381$) contained TGFB1 inducible genes, whose expression increased after *Hmga2*-KO, while TGFB1 treatment in *Hmga2*−/− MEF reduced their expression. We focused on cluster 1 ($n = 640$) for further analysis, which contained TGFB1 inducible genes in *Hmga2*-dependent manner. Cross-analysis of our RNA-seq after TGFB1 treatment (Fig. 8a and Supplementary Fig. 9a) with our ChIP-seq data (Fig. 3a and Supplementary Fig. 8f) confirmed the existence of three gene groups based on the position of the first nucleosome 3′ of the TSS containing pH2A.X, which we called position clusters 1 to 3 to differentiate them from the TGFB1 inducible clusters. Consistent with our previous results (Fig. 3b), the genes in the position clusters 1 to 3 displayed increasing basal transcription activity, whereby position cluster 1 has the lowest ($\bar{x} = 0.155$), position cluster 2 the middle ($\bar{x} = 0.662$) and position cluster 3 the highest ($\bar{x} = 2.766$) basal transcription activity in *Hmga2*+/+ MEF (Supplementary Fig. 9b). In addition, the position of the first nucleosome relative to the TSS also correlated with the strength of transcriptional activation induced by TGFB1 (Fig. 8b), where position cluster 1 showed the lowest ($\bar{x} = 29.55$), cluster 2 a medium ($\bar{x} = 40.23$) and cluster 3 the highest ($\bar{x} = 51.71$) transcriptional inducibility by TGFB1 in *Hmga2*+/+ MEF. Remarkably, *Hmga2*-KO reduced the inducibility of the genes after TGFB1 treatment in all three position clusters to 3.728 ($P = 2.18E\text{-}10$) in position cluster 1, 8.367 ($P = 1.17E\text{-}15$) in position cluster 2 and 5.328 ($P = 1.71E\text{-}12$) in position cluster 3. ChIP-seq analysis of pH2A.X levels was also performed using the same conditions as in our RNA-seq after TGFB1 treatment (Fig. 8c and Supplementary Fig. 9c). TGFB1 treatment increased pH2A.X levels from 2.23 to 2.937 ($P = 7.5E\text{-}3$) at the TSS of TGFB1 inducible cluster 1 genes in *Hmga2*+/+ MEF, whereas this effect was not observed in *Hmga2*−/− MEF, confirming the requirement of *Hmga2* for the effects induced by TGFB1.

Further, to determine the causal involvement of the FACT complex during TGFB1 induced transcriptional activation, we performed a series of experiments analyzing *Gata6*, *Mtor* and *Igf1* in *Hmga2*+/+ and *Hmga2*−/− MEF that were non-treated or treated with FACTin (Fig. 8d–f). TGFB1 treatment in *Hmga2*+/+ MEF increased the expression of *Gata6*, *Mtor* and *Igf1* (Fig. 8d) as well as the levels of pPol II and pH2A.X in their promoters (Fig. 8e), whereas 5mC levels were reduced (Fig. 8f). The effects induced by TGFB1 treatment were not observed in *Hmga2*−/− MEF confirming the requirement of *Hmga2*. Further, FACTin treatment counteracted the effects induced by TGFB1 in *Hmga2*+/+ MEF supporting the causal involvement of the FACT complex. Interestingly, WB analysis of protein extracts from *Hmga2*+/+ and *Hmga2*−/− MEF (Supplementary Fig. 9d) demonstrated that the effects observed after *Hmga2*- and FACT-LOF take place neither affecting total SMAD2/3 levels, nor changing their activation by TGFB1. Consistent with the mechanism of transcriptional regulation proposed here (Fig. 7b)

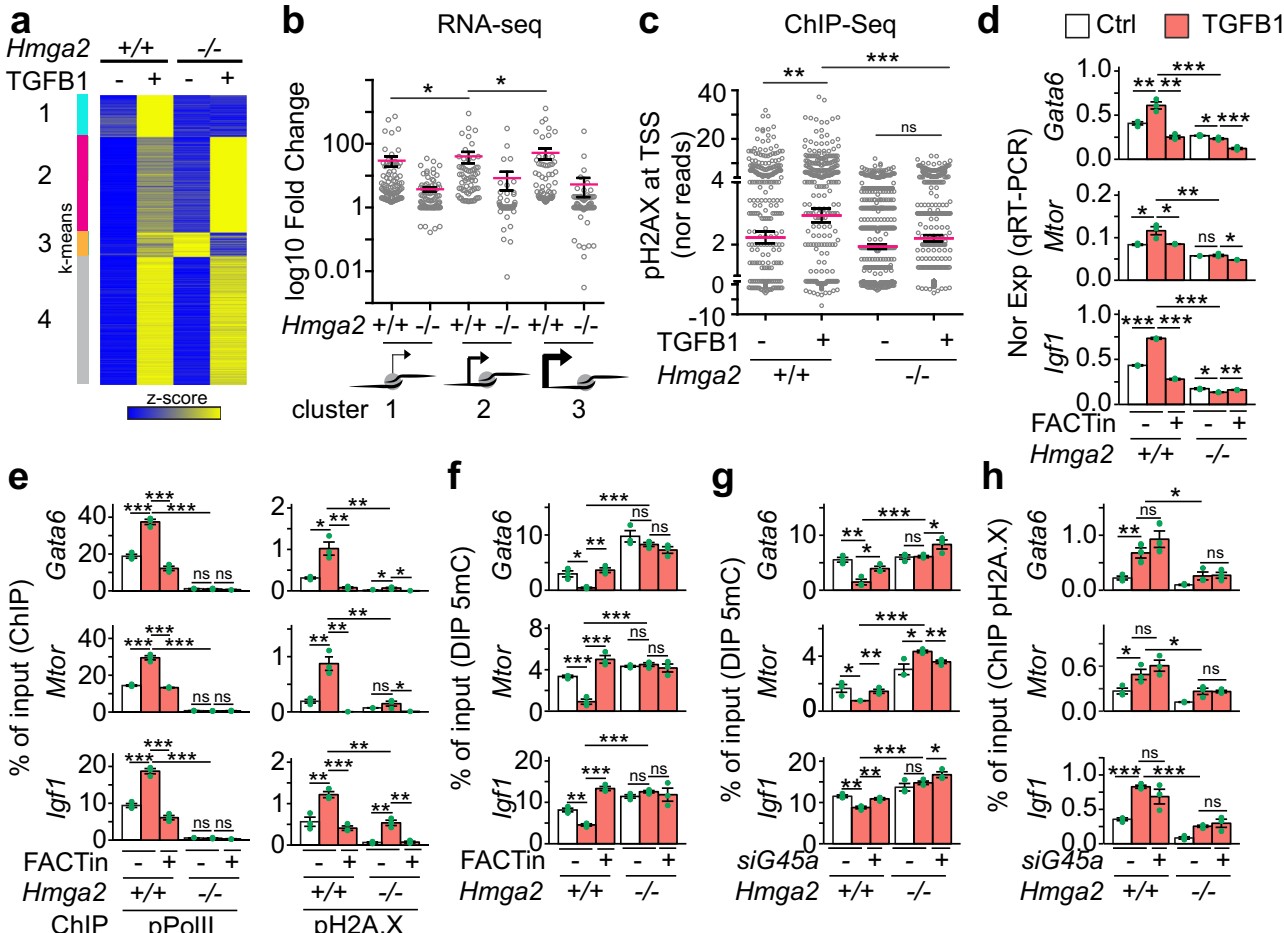

**Fig. 8 HMGA2-FACT-ATM-pH2A.X axis mediates TGFB1 induced transcription activation. a** Heat map showing RNA-seq-based expression analysis of TGFB1-inducible genes in *Hmga2*+/+ and *Hmga2*−/− MEF non-treated or treated with TGFB1. Data were normalized by Z score transformation and clustered using *k*-means algorithm. **b** Dot plot presenting the RNA-seq-based expression analysis of cluster 1 genes from A as log10 fold change between TGFB1-treated and non-treated MEF. Genes were grouped into the position clusters identified in Fig. 3a. Each dot represents the value of a single gene; *n* = 95 genes in position cluster 1; *n* = 64 genes in position cluster 2; *n* = 78 genes in position cluster 3; red line, average; error bars, s.e.m.; asterisks, *P*-values after one-tailed Mann–Whitney Test, \**P* ≤ 0.05. **c** Dot plot presenting ChIP-seq-based pH2A.X enrichment analysis in the top 15% candidates from Fig. 1c using chromatin from MEF treated as in a. For each gene, normalized reads in the TSS + 0.25 kb region were binned and the maximal value was plotted. Inputs were subtracted from the corresponding samples. Red line, average; *n* = 640 genes; error bars, s.e.m.; asterisks, *P*-values after one-tailed Mann–Whitney test, \*\*\**P* ≤ 0.001; \*\**P* ≤ 0.01; ns, non-significant. **d** qRT-PCR-base expression analysis of HMGA2 target genes in *Hmga2*+/+, *Hmag2*−/− MEF that were non-treated (Ctrl) or treated with TGFB1 and FACT inhibitor (FACTin; CBLC000 trifluoroacetate) as indicated. **e** ChIP-based promoter analysis of selected HMGA2 target genes using the indicated antibodies and chromatin from MEF treated as in **d**. **f** DIP-based promoter analysis of selected HMGA2 target genes using antibodies specific for 5-methylcytosine (5mC) and genomic DNA from MEF treated as in **d**. **g** DIP-based promoter analysis of selected HMGA2 target genes using the indicated antibodies and genomic DNA from *Hmag2*+/+ or *Hmag2*−/− MEF that were transfected with control (−) or *Gadd45a*-specific small interfering RNA (siRNA) as indicated. **h** ChIP-based promoter analysis of selected HMGA2 target genes using pH2A.X-specific antibodies and chromatin from MEF treated as in g. In all bar plots, data are shown as means ± s.e.m. (*n* = 3 biologically independent experiments); asterisks, *P*-values after two-tailed *t*-test, \*\*\**P* ≤ 0.001; \*\**P* ≤ 0.01; \**P* ≤ 0.05; ns, non-significant. See also Supplementary Fig. 9. Source data are provided as a Source Data files 01 and 04.

and the results in Fig. 7e, f, siRNA-mediated *Gadd45a*-LOF counteracted the reducing effect of TGFB1 on 5mC levels in *Hmga2* + /+ MEF (Fig. 8g) without affecting the increasing effect on pH2A.X levels (Fig. 8h), thereby confirming that GADD45A acts downstream of the HMGA2-FACT-ATM-pH2A.X axis.

**Inhibition of the FACT complex counteracts fibrosis hallmarks in IPF.** The clinical potential of the here proposed mechanism of transcription regulation (Fig. 7b) was approached by placing it into the context of the most common interstitial lung disease, IPF, in which TGFB signaling plays a key role[44]. RNA-seq in primary

human lung fibroblasts (hLF) isolated from control (*n* = 3) and IPF (*n* = 3) patients revealed increased expression levels of *HMGA2*, *SUPT16H* and *SSRP1* in IPF patients when compared to control donors (Supplementary Fig. 10a). Further, cross-analysis of RNA-seq in primary hLF isolated from control and IPF patients[44] with RNA-seq in *Hmga2* + /+ MEF that were non-treated or treated with TGFB (Fig. 9a) allowed us to identify 923 orthologue genes that were at least 1.5 fold significantly increased (P ≤ 0.05) in IPF hLF and in TGFB treated MEF when compared to the corresponding control cells. Gene set enrichment analysis (GSEA)[45] based on normalized enrichment scores (NSE) revealed significant enrichment of orthologue genes related to EMT

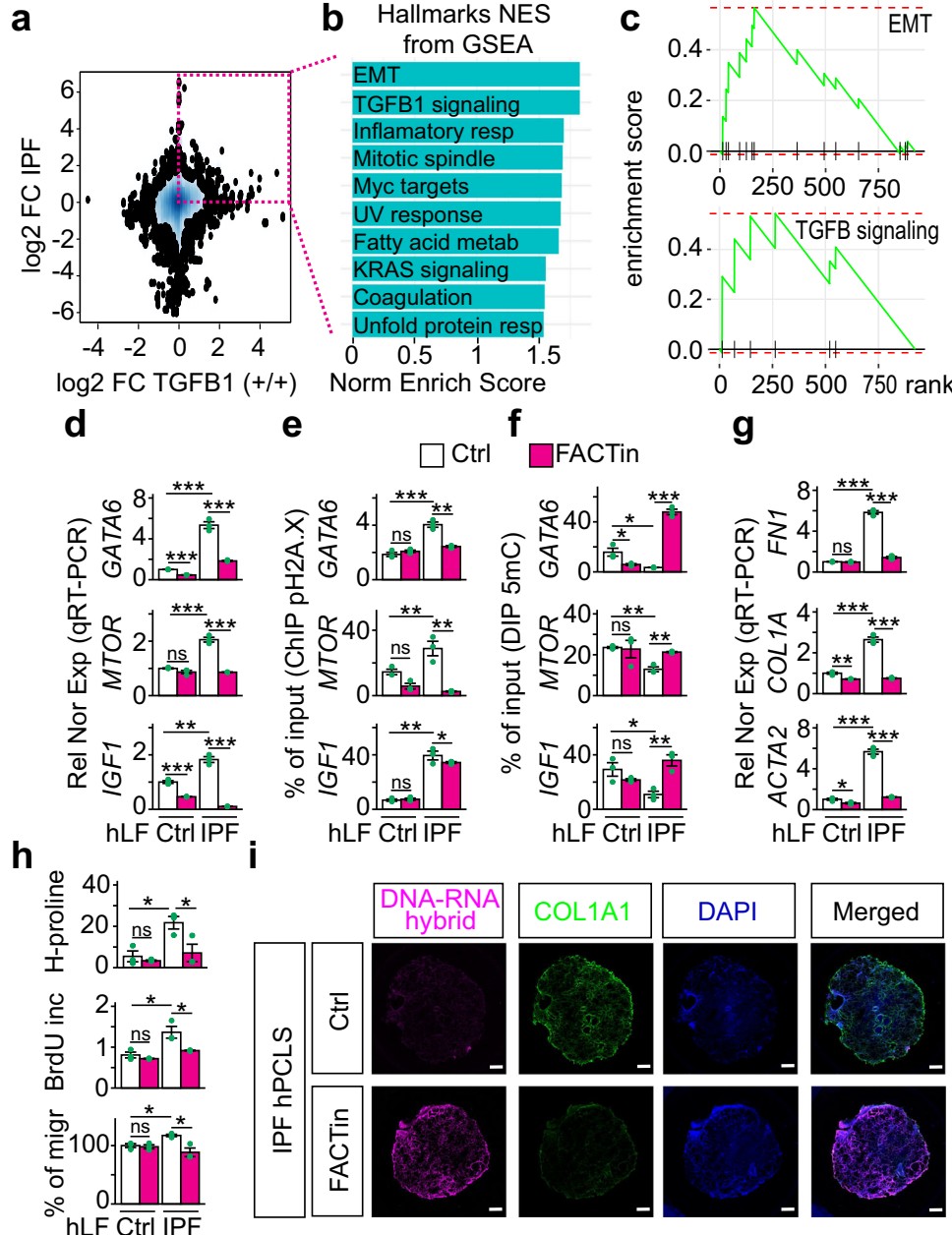

**Fig. 9 Inhibition of the FACT complex counteracts fibrosis hallmarks in IPF. a** RNA-seq-based comparison of gene expression in IPF and after TGFB1 treatment. 2D Kernel Density plot representing the log2 fold change between gene expression in primary human lung fibroblasts (hLF) from IPF patients vs. control donors on the *y*-axis and log2 fold change between gene expression in *Hmga2+/+* MEF treated with TGFB1 vs. non-treated on the *x*-axis. Square, genes with log2 FC > 0.58 and $P \leq 0.05$ in both, hLF IPF and TGFB1-treated MEF. *P*-values after Wald test. **b** Gene set enrichment analysis (GSEA) using the normalized enrichment scores (NES) of genes inside the square in a. EMT, epithelial-mesenchymal transition; resp, response. **c** GSEA line profile of the top two enriched pathways in **b**. **d** qRT-PCR-based expression analysis of selected HMGA2 target genes in hLF from control donors (Ctrl) or IPF patients that were non-treated (Ctrl) or treated FACT inhibitor (FACTin; CBLC000 trifluoroacetate) as indicated. **e** ChIP-based promoter analysis of selected HMGA2 target genes using pH2A.X-specific antibodies and chromatin from hLF treated as in **d**. **f** DIP-based promoter analysis of selected HMGA2 target genes using 5mC-specific antibodies and genomic DNA from hLF treated as in **d**. **g** qRT-PCR-based expression analysis of fibrotic markers in hLF treated as in **d**. FN1, fibronectin; COL1A1, collagen; ACTA2, smooth muscle actin alpha 2. **h** Functional assays for IPF hallmarks in Ctrl or IPF hLF treated as in **d**. Top, hydroxyproline assay for collagen content. Middle, proliferation assay by BrdU incorporation. Bottom, Transwell invasion assay. **i** Representative pictures from confocal microscopy after immunostaining using the antibody S9.6 or COL1A1-specific antibody in human precision-cut lung slices (hPCLS) from IPF patients (*n* = 3 biologically independent experiments). The hPCLS were treated as in d. DAPI, nucleus. Scale bars, 500 μm. In all bar plots, data are shown as means ± s.e.m. (*n* = 3 biologically independent experiments); asterisks, *P*-values after tow-tailed *t*-test, \*\*\**P* ≤ 0.001; \*\**P* ≤ 0.01; \**P* ≤ 0.05; ns, non-significant. See also Supplementary Fig. 10. Source data are provided as a Source Data files 01 and 04.

($P = 6.3E-3$), TGFB1 signaling pathway ($P = 0.012$), inflammatory response ($P = 0.033$), MYC target genes ($P = 0.011$), UV response $P = 0.031$), fatty acid metabolism ($P = 0.029$), among others (Fig. 9b). In addition, graphical representation of the enrichment profile showed high enrichment scores (ES) for EMT (ES = 0.515) and TGFB1 signaling pathway (ES = 0.706) as the top two items of the ranked list (Fig. 9c). To determine the role of the HMGA2-FACT-ATM-pH2A.X axis in IPF we analyzed *GATA6*, *MTOR* and *IGF1* in Ctrl and IPF hLF (Fig. 9d–f). Correlating with the results obtained in MEF after TGFB1 treatment (Fig. 8d–f), we detected in IPF hLF increased expression of *GATA6*, *MTOR* and *IGF1* (Fig. 9d), as well as increased levels of pH2A.X and H2A.X in their promoters (Fig. 9e and Supplementary 10b), whereas 5mC levels were reduced (Fig. 9f). Strikingly, FACTin treatment counteracted the effects observed in IPF hLF, supporting the involvement of the HMGA2-FACT-ATM-pH2A.X axis in this interstitial lung disease.

To test this hypothesis, we monitored various hallmarks of fibrosis in Ctrl and IPF hLF, such as expression of fibrotic markers by qRT-PCR (Fig. 9g), levels of ECM proteins by Hydroxyproline and Sircol assays (Fig. 9h, top and Supplementary 10c), cell proliferation by bromodeoxyuridine (BrdU) incorporation assay (Fig. 9h, middle) and cell migration by Transwell invasion assay followed by hematoxylin and eosin (H&E) staining (Fig. 9h, bottom). Remarkably, FACTin treatment of IPF hLF significantly reduced all hallmarks of fibrosis analyzed, thereby suggesting the use of FACTin for therapeutic approaches against IPF. Moreover, our in vitro findings in primary hLF were also confirmed ex vivo using human precision-cut lung slices (hPCLS) from 3 different IPF patients (Fig. 9i and Supplementary Fig. 10d–g). FACTin treatment of IPF hPCLS reduced the levels of the fibrotic markers COL1A1, FN1, smooth muscle actin alpha 2 (ACTA2), the mesenchymal marker vimentin (VIM), as well as HMGA2 and pH2A.X. In contrast, the levels of DNA-RNA hybrids were increased after FACTin treatment.

## Discussion

Here, we uncovered a mechanism of transcription initiation of TGFB1-responsive genes mediated by the HMGA2-FACT-ATM-pH2A.X axis. The lyase activity of HMGA2 induces DNA nicks at the TSS, which are required by the FACT complex to incorporate nucleosomes containing H2A.X at specific positions relative to the TSS. The position of the first nucleosome containing H2A.X not only correlates with the basal transcription activity of the corresponding genes, but also with the strength of their inducibility after TGFB1 treatment. Further, ATM-mediated phosphorylation of H2A.X at S139 is required for repair-mediated DNA demethylation and transcriptional activation. Our data support a sequential order of events, in which specific positioning of nucleosomes containing the classical DNA damage marker pH2A.X precedes DNA demethylation and transcription initiation, thereby supporting the hypothesis that chromatin opening involves intermediates with DNA breaks that require mechanisms of DNA repair that ensure the integrity of the genome.

### Biological and clinical relevance of the HMGA2-FACT-ATM-pH2A.X axis. We demonstrated the biological relevance of our data within the context of TGFB1 signaling (Fig. 8 and Supplementary Fig. 9). TGFB1 treatment induced promoter specific increase of pH2A.X and pPol II, whereas 5mC levels were decreased, resulting in transcription activation in *Hmga2*- and FACT-dependent manner (Fig. 8a–f). Interestingly, *Gadd45a*-LOF interfered with the 5mC decrease (Fig. 8g), without affecting pH2A.X levels (Fig. 8h), supporting the sequential order of events proposed here (Fig. 7b), in which GADD45A acts downstream of

the HMGA2-FACT-ATM-pH2A.X axis. Consistent with our findings, Thillainadesan and colleagues reported TGFB induced active DNA demethylation and expression of the *p15^{ink4b}* tumor suppressor gene[17]. While published reports showed the effect of TGFB on specific genes[10,17,46], in this report we demonstrated the genome wide effect of TGFB treatment affecting the global nuclear architecture and strongly suggesting future NGS studies. Following a similar line of ideas, Negreros and colleagues recently reported genome wide changes on DNA methylation induced by TGFB1[47]. The translational potential of our work was demonstrated within the context of IPF (Fig. 9 and Supplementary Fig. 10), in which TGFB1 signaling plays an important role. Inhibition of the HMGA2-FACT-ATM-pH2A.X axis reduced all fibrotic hallmarks in vitro (using primary hLF) and ex vivo (using hPCLS). Interestingly, the FACT complex is a potential marker of aggressive cancers with low survival rates[48] and FACTin is being tested in a clinical trial for cancer treatment (ClinicalTrials.gov Identifier: NCT01905228, NCT02931110). Our work provides the molecular basis for future studies developing therapies against IPF using FACTin.

## Methods

**Key resources.** Please see Supplementary Table 2 containing the key resources used in this study.

**Study design.** This study was performed according to the principles set out in the WMA Declaration of Helsinki; the underlying protocols were approved by the ethics committee of Medicine Faculty of the Justus Liebig University in Giessen, Germany (AZ.111/08-eurIPFreg) and the Hannover Medical School (no. 2701-2015). In this line, all patient and control materials were obtained through the UGMLC Giessen Biobank (member of the DZL Platform Biobanking) and the Biobank from the Institute for Pathology of the Hannover Medical School as part of the BREATH Research Network. We used anonymized patient material.

**Cell culture.** All studies were done on immortalized MEF cultivated for less than twenty passages. *Hmga2* wild type (+/+) and knockout (−/−) primary mouse embryonic fibroblast (MEF) were isolated from mouse embryos at embryonic day E15.5 and subsequently immortalized using simian virus (SV) 40[10]. MEF and Human embryonic kidney cell HEK293T (ATCC, CRL-11268) were cultured at 37 °C in 5% $CO_2$ in DMEM medium with 4.5 g/l glucose, 10% FCS 4 mM L-Glutamine, 1 mM Pyruvate, 100 U/ml penicillin and 100 U/ml streptomycin. Mouse lung epithelial cells (MLE-12, ATCC CRL-2110) were cultured in Dulbecco's Modified Eagle Medium: Ham's F-12 Nutrient Mixture (5% FCS, 100 U/ml penicillin and 100 U/ml streptomycin) at 37 °C in 5% $CO_2$. Primary fibroblast from Ctrl and IPF patients were cultured in complete MCDB131 medium (8% FCS, 1% L-glutamine, penicillin 100 U/ml, streptomycin 0.1 mg/ml, EGF 0.5 ng/ml, bFGF 2 ng/ml, and insulin 5 μg/ml)) at 37 °C in 5% $CO_2$. Because of the concern that the phenotype of the cells is altered at higher passage, cells between passages 4 and 6 were utilized in the experiments described here. All cells were washed with 1x PBS, trypsinized with 0.25% (w/v) trypsin and subcultivated at the ratio of 1:5 to 1:10.

**Cell treatments, transfections and siRNA-mediated knockdown.** MEF were treated with 1 μg/ml doxycycline or DMSO (used as solvent for doxycycline) for 4, 6 or 24 h to induce the expression of transgenes. Initial FACT complex and ATM kinase inhibition was performed with 5 μM CBLC000 trifluoroacetate (FACTin, Sigma Aldrich) for 2 h or 1 μM KU-55933 (ATMi, Calbiochem) for 6 h, respectively. MEF were transiently transfected either with 100 nM siCtrl (negative control; AM4611, Ambion) or *siGadd45a* (siG45; AM16708, Ambion) for 48 h. TGFB1 signaling was induced after a 16 h starvation (cell culture medium supplemented with 1% FCS) with 10 ng/ml human recombinant TGFβ1 (Sigma Aldrich) and chromatin changes were assayed after 3 h and gene expression alterations after 24 h incubations. For IPF resolution experiments, primary hLF were treated with 5 μM FACTin for 12 h.

**Bacterial culture.** For cloning experiments, chemically competent *E. coli* TOP10 (Thermo Fisher Scientific) were used for plasmid transformation. TOP10 strains were grown in Luria broth (LB) at 37 °C with shaking at 180 rpm on LB agar at 37 °C overnight.

**Chromatin immunoprecipitation.** Cells were cross-linked with 1% formaldehyde for 10 min at room temperature (RT) and quenched on ice with 125 mM glycine for 5 min. Media was aspirated and cells were washed 3 times with ice cold PBS. Cells were lifted in lysis buffer (50 mM Tris.HCl pH 8.0, 2 mM EDTA pH 8.0, 0.1%

NP40, 10% glycerol) and kept on ice for additional 5 min. Nuclei were collected by centrifugation at 300 × g for 10 min at 4 °C followed by resuspension in nuclear resuspension buffer (50 mM Tris.HCl pH 8.0, 5 mM EDTA pH 8.0, 1 % SDS). Nuclei were sonicated with Diagenode Bioruptor to an average DNA length of 300–600 bp. After centrifugation, the soluble chromatin was diluted 1:20 with DB dilution buffer (50 mM Tris.HCl pH 8.0, 5 mM EDTA pH 8.0, 0.5% NP40, 0.2 M NaCl) and 25 µg chromatin was incubated with 3 µg of antibodies specific for H3 (Abcam), H4 (Abcam), H2A (Abcam), H2B (Abcam), pH2A.X (Millipore), H2A.X (Millipore), HMGA2 (Abcam), pPol II (Abcam), Pol II (Abcam), SSRP1 (Biolegend), GADD45A (Santacruz), TET1 (Active Motif), SPT16 (Cell Signaling) and IgG as a control (Santa Cruz Biotechnology)(see Supplementary Table 2 for antibody details). Sepharose G beads were preblocked with 1% BSA and incubated for 2 h to precipitate chromatin. Beads were washed sequentially with low salt washing buffer (20 mM Tris.HCl pH 8.0, 2 mM EDTA pH 8.0, 1% NP40, 0.1% SDS, 0.15 M NaCl), high salt washing buffer (20 mM Tris.HCl pH 8.0, 2 mM EDTA pH 8.0, 1% NP40, 0.1 % SDS, 0.5 M NaCl), LiCl washing buffer (10 mM Tris.HCl pH 8.0, 1 mM EDTA pH 8.0, 1% NP40, 1% Na-deoxycholate, 0.25 M LiCl) and twice with TE buffer (10 mM Tris.HCl pH 8.0, 1 mM EDTA pH 8.0). Precipitated chromatin was reverse-crosslinked by incubating with 100 µl of 1% SDS, 0.1 M NaHCO3 and 0.5 µl RNase A (1 mg/ml) for 30 min at 37 °C. Then 2.5 µl of Proteinase K (10 mg/mL) was added for 2 h at 56 °C. Lastly, NaCl was adjusted to a final concentration of 0.2 M and mix was incubated for 16 h at 65°C. Immunoprecipitated chromatin was purified using the QIAquick PCR purification kit (Qiagen) and subjected to qPCR or next-generation sequencing. For qPCR, the percentage of input was calculated after subtracting the IgG background, if not stated elsewhere.

### ChIP sequencing and data analysis in *Hmga2*+/+ and *Hmga2*−/− MEF.
Libraries were prepared according to Illumina's instructions accompanying the Ovation Ultra Low Kit. Single-end sequencing was performed on an Illumina HiSeq2500 machine at the Max Planck-Genome-Centre Cologne. Raw reads were visualized by FastQC (https://www.bioinformatics.babraham.ac.uk/projects/fastqc/) to determine the quality of the sequencing. Trimming was performed using trimmomatic[49] with the following parameters LEADING:3 TRAILING:3 SLIDINGWINDOW:4:15 MIN-LEN:50 CROP:63 HEADCROP:13. High quality reads were mapped by using Bowtie2[50] to mouse genome mm10. ChIP-seq data were represented as aggregate plot or heat maps using deeptools[51] following their instructions. Bam-files were converted to Bed-files using Bedtools' bamToBed (https://bedtools.readthedocs.io/en/latest/) command. Genome browser snapshots were created with Homer using makeTagDirectory (http://homer.ucsd.edu/homer/ngs/tagDir.html) and makeUCSCfile (http://homer.ucsd.edu/homer/ngs/ucsc.html). Reads were normalized to 30 million reads. Peak calling was performed by using model-based analysis for ChIP-seq (MACS)[52] with a cut-off of $p < 0.01$ and the following parameters: --nonmodel, –shift size 30, and an effective genome size -g of 1.87e9. Peaks were annotated by using annotePeaks.pl for mm10 from Homer (http://homer.ucsd.edu/homer/ngs/annotation.html). From the 200,051 peaks found for pH2A.X in *Hmga2*+/+ MEF, 3,935 peaks were annotated in the promoter-TSS region and were used for further analysis.

### Identification of position clusters and analysis of inducibility.
The enrichment of pH2A.X at TSS plus 250 bp downstream in the top 15% candidates was clustered using the *k*-means algorithm implemented in deeptools' plotHeatmap command. From the 3 position clusters identified, genes in *Hmga2*+/+ MEF with a FC more than 1.5 to *Hmga2*−/− were selected. For expression analysis, RPKMs equal or higher than 10 were considered as outliers (Number of outliers: C1, 5 genes; C2, 2 genes; and C3, 0 genes).

### RNA isolation, reverse transcription, quantitative PCR.
Total RNA was isolated with Trizol (Invitrogen) and quantified using a Nanodrop Spectrophotometer (ThermoFisher Scientific, Germany). Synthesis of cDNA was performed using 0.5-1 µg total RNA and the High Capacity cDNA Reverse Transcription kit (Applied Biosystems). Quantitative real-time PCR reactions were performed using SYBR® Green on the Step One plus Real-time PCR system (Applied Biosystems). The housekeeping genes *Tuba1a* and *HPRT1* were used to normalize gene expression[53]. Primer pairs used for gene expression analysis are described in Supplementary Information, Supplementary Table 1.

### Native chromatin fractionation and sucrose gradient ultracentrifugation.
Two 15 cm dishes with MEF were washed with 1× PBS and pellets were resuspended in 1 ml of lysis buffer (10 mM HEPES pH 7.4, 10 mM KCl, 0.05% NP-40, 1 mM DTT, 25 mM NaF, 0.5 mM Na₃VO₄, 40 µg/ml phenylmethylsulfonyl fluoride and protease inhibitor). After incubating 20 min on ice, cells were spun down at 300 × g at 4 °C for 10 min. The nuclei were washed once in lysis buffer and were then resuspended in 2 volumes of Low Salt Buffer (10 mM Tris-HCl pH 7.4, 0.2 mM MgCl₂ supplemented with protease and phosphatase inhibitors) including 1% Triton-X100. After 15 min incubation on ice, cells were spun down at 300 × g at 4 °C for 10 min. The pellet was washed in 1 ml MNase digestion buffer (10 mM Tris-HCl, 25 mM NaCl, 1 mM CaCl2, 1 mM DTT, 25 mM NaF, 0.5 mM Na₃VO₄, 40 µg/ml phenylmethylsulfonyl fluoride and protease inhibitor) and resuspended again in 1 ml MNase digestion buffer with 1,250 Units MNase (NEB Biolabs). Chromatin-MNase mix was incubated at 37 °C for 30 min. MNase reaction was

stopped by adding 50 mM EDTA. Samples were sonicated for 30 sec on / 30 sec off using the Bioruptor with high amplitude. Chromatin was spun down at 18,407 ×g at 4 °C for 10 min and the supernatant was used for ultracentrifugation. Sucrose gradients (5% to 40%) were prepared in 1,800 µl low salt buffer (10 mM NaCl, 10 mM Tris-HCl pH7.4, 0.2 mM EDTA, 0.2 mM DTT, 20 mM NaF, 20 mM Na₃VO₄, 40 µg/ml phenylmethylsulfonyl fluoride and protease inhibitor) in polyallomer centrifuge tube (Beckman). Fragmented native chromatin was loaded on top of the 9 ml 5% to 40% sucrose gradients and centrifuged for 16 h and 30 min at 168,544 ×g in a SW50.1 ultracentrifuge rotor (Beckman Coulter). Following centrifugation, 11 fractions (1000 µl each) were collected manually from the bottom of the tubes. Later, these fractions were used for western blot, mass spectrometry and NGS.

Western blotting was performed using standard methods and antibodies specific for HMGA2 (Abcam; 1:1,000 dilution), pPol II (Abcam; 1:1,000 dilution), total Pol II (Abcam; 1:1,000 dilution), SUPT16 (Cell Signaling; 1:1,000 dilution), SSRP1 (Biolegend; 1:1,000 dilution), pH2A.X (Millipore; 1:1,000 dilution), H2A.X (Abcam; 1:1,000 dilution), H2A (Abcam; 1:1,000 dilution), H2B (Abcam; 1:1,000 dilution) and H3 (Abcam; 1:5,000 dilution) were used (see Supplementary Table 2 for antibody details). Immunoreactive proteins were visualized with the corresponding HRP-conjugated secondary antibodies (Jackson; 1:10,000 dilution) using the WesternBright ECL detection solutions (Biozym). Signals were detected and analyzed with Luminescent Image Analyzer (Las 4000, Fujifilm).

### Mass spectrometry: sample preparation, methods and data analysis.
Proteins were methanol/chloroform precipitated from sucrose gradient fractions and dried pellets reconstituted in 8 M urea[54]. Per fraction, 140 µg of protein (according to the 660 nm protein assay, Pierce), were subjected to in-solution digest using protein to enzyme ratios of 1:100 and 1:50 for Lys-C (Wako Chemicals GmbH) and trypsin (Serva), respectively[55]. The resulting peptide mixture was desalted and concentrated using Oligo R3 (Thermo Fisher Scientific) extraction[56]. Peptides (5 µg according to the quantitative fluorimetric peptide assay, Pierce), were subsequently labeled using 6-plex tandem mass tags (Thermo Fisher Scientific) following the manufacturer's protocol but employing a reagent to peptide ratio of four. Labeling channels were used for fractions from a replicate sucrose gradient as well as an internal standard sample consisting from an analogously treated mix of all replicate gradient input samples. After validation of labeling efficiency by liquid chromatography-tandem mass spectrometry (LC-MS2), samples were mixed by equal protein amount and 4 µg total peptides purified as well as concentrated using STAGE tips[57]. The subsequent LC-MS2 analysis of 50% of that peptide material used an in-house packed 70 µm ID, 15 cm reverse phase column emitter (ReproSil-Pur 120 C18-AQ, 1.9 µm, Dr. Maisch GmbH) with a buffer system comprising solvent A (5% acetonitrile, 0.1% formic acid) and solvent B (80% acetonitrile, 0.1% formic acid). Relevant instrumentation parameters are extracted using MARMo-SET and included in the supplementary material as Source Data File 03[58]. Peptide/protein group identification and quantitation was performed using the MaxQuant suite of algorithms (v. 1.6.5.0) against the mouse uniprot database (canonical and isoforms; downloaded on 2019/01/23; 86695 entries)[59,60]. For downstream analysis, intensities of fractions were divided by their corresponding inputs and samples that were divided by zero were set to 0.1.

### MNase-sequencing and data analysis.
For DNA purification from sucrose ultracentrifugation fractions, 200 µl of fractions were resuspended with 200 µl 1× PBS and incubated with 0.5 µl RNase A (10 mg/ml, Sigma Aldrich) for 15 min at 37 °C. Samples were resuspended with 400 µl Ultra-pure phenol:chloroform (Invitrogen) and incubated for 5 min at RT. After centrifugation for 5 min at 18,407g at 4 °C, the clear phase containing DNA was transferred into a fresh tube. 40 µl of 3 M sodium acetate pH 4.9 and 1 ml of ethanol were added to the samples and DNA was precipitated for 30 min at −80 °C. DNA-mix was spun down for 30 min at 18,407g at 4 °C and the DNA pellets were washed with 70% ice cold ethanol. After centrifugation for 15 min at 18,407g at 4 °C, the pellets were dried at RT and further resuspended in 50 µl of nuclease-free water and heated for 15 min at 37 °C. DNA was analyzed by agarose gel electrophoresis or NGS.

For sequencing, purified DNA was quantified by Qubit dsDNA HS Assay Kit (Thermo Fisher Scientific). 10 ng DNA was used as input for TruSeq ChIP Library Preparation Kit (Illumina) with following modifications. Instead of gel-based size selection before final PCR step, libraries were size selected by SPRI-bead based approach after final PCR with 18 cycles. In detail, samples were first cleaned up by 1x bead:DNA ratio to eliminate residuals from PCR reaction, followed by 2-sided-bead cleanup step with initially 0.6x bead:DNA ratio to exclude larger fragments. Supernatant was transferred to new tube and incubated with additional beads in 0.2x bead:DNA ratio for eliminating smaller fragments, like adapter and primer dimers. Bound DNA samples were washed with 80% ethanol, dried and resuspended in TE buffer. Library integrity was verified with LabChip Gx Touch 24 (Perkin Elmer). Sequencing was performed on the NextSeq500 instrument (Illumina) using v2 chemistry with 2x38bp paired setup. Raw reads were visualized by FastQC to determine the quality of the sequencing. Trimming was performed using trimmomatic with the following parameters LEADING:3 TRAILING:3 SLIDINGWINDOW:4:15 HEADCROP:5, MINLEN:15. High quality reads were mapped by using with bowtie2 to mouse genome mm10. For downstream analysis, fragments between 100 and 200 bp were selected and the reads were centered. Reads were normalized by an RPKM measure and represented as log2 enrichment

over the corresponding inputs. To avoid division through zero, zero counts were pseudo-counted as "1".

**Generation of HMGA2 lyase mutant, doxycycline inducible MEF and stable knockdown of *Hmga2* in MLE-12 cells**. The lyase deficient mutant of HMGA2 was generated by sequential site-directed mutagenesis (QuikChange II Site-Directed Mutagenesis Kit, Agilent) of the first three arginines in the third AT-Hook domain. Mutagenesis primers for *Hmga2* are listed in Supplementary Table 1. All constructs were sequence verified.

Replication-deficient lentiviruses containing doxycycline inducible Ctrl (Empty), *Hmga2* WT-myc-his and *Hmga2* RΔA-myc-his were produced by transient transfection of pCMVR8.74 (Addgene: #22036), pMD2.G (Addgene: #12259) and transfer plasmid into HEK293T cells in a 6-well plate. Viral supernatants were collected after 48 h, spun down at 3,488 ×g for 20 min, and then used to transduce immortalized MEF in the presence of polybrene (10 μg/ml, Sigma). GOF experiments for GADD45A require a high transfection efficiency which were not obtained by standard transfection protocols in MEF. Therefore, *Hmga2* was knocked down by shRNA in MLE-12 cells which allowed transient transfection of a *GADD45A* expression construct. For a stable KD of *Hmga2* in MLE-12 cells, lentiviruses for pLKO-*scrambled* or pLKO-*shHmga2* were produced and transduced as described before. Forty-eight h later, MEF and MLE-12 cells were selected by stepwise increase with 1.5 to 3.0 or 4.0 μg/ml puromycin respectively and the pooled populations were used for various experiments.

**Co-immunoprecipitation (Co-IP) and Western blot**. MEF were washed three times in cold 1× PBS and scraped in 10 ml of lysis buffer (10 mM HEPES pH 7.4, 10 mM KCl, 0.05% NP-40, 1 mM DTT, 25 mM NaF, 0.5 mM Na$_3$VO$_4$, 40 μg/ml phenylmethylsulfonyl fluoride and protease inhibitor). Cells were incubated on ice for 20 min and then spun down at 300g for 10 min at 4 °C. Nuclear cell pellets were resuspended in 300 μl Co-IP buffer (50 mM Tris-HCl pH 7.4, 170 mM NaCl, 20% glycerol, 15 mM EDTA, 0.1% (v/v) Triton X-100, 20 mM NaF, 20 mM Na$_3$VO$_4$, 40 μg/ml phenylmethylsulfonyl fluoride and protease inhibitor) and were sonicated 5 times using the Bioruptor (30 sec on, 30 sec off). Soluble chromatin and proteins were collected after centrifugation at 10,000 × g at 4 °C for 10 min. Precleared nuclear protein lysates (500 μg) were incubated with 20 μl of anti-c-MYC tag antibody coupled to magnetic beads (Thermo Scientific). After two hours, beads were collected and washed 4 times with 500 μl ice-cold washing buffer (50 mM Tris-HCl pH 7.4, 170 mM NaCl, 15 mM EDTA, 0.4% (v/v) Triton X-100, 20 mM imidazole, 20 mM NaF, 20 mM Na$_3$VO$_4$, 40 μg/ml phenylmethylsulfonyl fluoride and protease inhibitor). Proteins were eluted in 30 μl 2x SDS sample loading buffer while boiling at 95 °C for 5 min. Beads were removed and protein eluates were loaded on SDS-PAGE for western blot analysis. Western blotting was performed using standard methods and antibodies specific for SUPT16 (Cell Signaling; 1:1000 dilution), SSRP1 (Biolegend; 1:1000 dilution), H2A.X (Millipore; 1:1000 dilution) and HIS-tag (Abcam; 1:1000 dilution) were used. Immunoreactive proteins were visualized with the corresponding HRP-conjugated secondary antibodies (Jackson; 1:10,000 dilution) using the WesternBright ECL detection solutions (Biozym). Signals were detected and analyzed with Luminescent Image Analyzer (Las 4000, Fujifilm). Protein concentrations were determined using Bradford kit (Pierce).

**Preparation of nuclear, chromatin and nucleoplasm extracts**. Cells were washed with 1× PBS and pellets were resuspended in 2 volumes of lysis buffer (10 mM HEPES pH 7.4, 10 mM KCl, 0.05% NP-40, 1 mM DTT, 25 mM NaF, 0.5 mM Na$_3$VO$_4$, 40 μg/ml phenylmethylsulfonyl fluoride and protease inhibitor (Calbiochem)). After 20 min incubation on ice, cells were spun down at 300 × g at 4 °C for 10 min. The supernatant was removed and nuclei were resuspended in whole cell lysate buffer (1% SDS, 10 mM EDTA pH 8, 50 mM Tris-HCl pH 8, 0.05% NP-40, 1 mM DTT, 25 mM NaF, 0.5 mM Na$_3$VO$_4$, 40 μg/ml phenylmethylsulfonyl fluoride and protease inhibitor). To obtain soluble nuclear proteins, resuspended nuclei were sonicated 5 times 30 s on followed by 30 sec off using the Bioruptor (Diagenode). Insoluble proteins were removed by centrifugation for 10 min at 18,407g at 4 °C.

For chromatin and nucleoplasm preparations, the protocol was adapted with minor modifications from[61]. Cells were washed with 1 x PBS and pellets were resuspended in 2 volumes of lysis buffer (10 mM HEPES pH 7.4, 10 mM KCl, 0.05% NP-40, 1 mM DTT, 25 mM NaF, 0.5 mM Na$_3$VO$_4$, 40 μg/ml phenylmethylsulfonyl fluoride and protease inhibitor). After 20 min incubation on ice, cells were spun down at 300g at 4 °C for 10 min. The supernatant contains the cytoplasmic proteins. The nuclei were washed once in lysis buffer and were then resuspended in 2 volumes of Low Salt Buffer (10 mM Tris-HCl pH7.4, 0.2 mM MgCl$_2$ supplemented with protease and phosphatase inhibitors including 1% Triton-X100). After 15 min incubation on ice, cells were spun down at 300g at 4 °C for 10 min. The supernatant contains the nucleoplasm proteins and the pellet contains the chromatin. Pellets were resuspended in 2 volumes of 0.2 N HCl and incubated on ice for 20 min. Resuspended pellets were spun down at 18,407g at 4 °C for 10 min and the supernatant containing acid soluble proteins was neutralized with the same volume of 1 M Tris-HCl pH 8.0. Western blotting was performed using standard methods and antibodies specific for SUPT16 (Cell Signaling; 1:1,000 dilution), SSRP1 (Biolegend; 1:1000 dilution), HMGA2 (Abcam; 1:1000 dilution), H3 (Abcam; 1:5,000 dilution) and AKT (Cell signaling; 1:1000

dilution) were used. Immunoreactive proteins were visualized with the corresponding HRP-conjugated secondary antibodies (Jackson; 1:10,000 dilution) using the WesternBright ECL detection solutions (Biozym). Signals were detected and analyzed with Luminescent Image Analyzer (Las 4000, Fujifilm). Protein concentrations were determined using Bradford kit (Pierce).

**DNA-RNA hybrid immunoprecipitation (DRIP)-qPCR**. Total nucleic acids were extracted from MEF by SDS/Proteinase K treatment at 37 °C followed by phenol-chloroform extraction and ethanol precipitation. Free RNA was removed by RNAse A treatment. DNA was fragmented overnight using HindIII, EcoRI, BsrGI, XbaI, and SspI and pretreated, or not, with 40 U RNase H1 (NEB, 5000 U/ml). For DRIP, R-loops were immunoprecipitated using 6 μg DNA-RNA hybrids antibody (Kerafast, Cat. ENH001) per 10 μg of digested DNA in 500 μl IP buffer. Bound R-loops were recovered by addition of 50 μl pre-blocked dynabeads protein A magnetic beads (Thermo Fisher Scientific) followed by two washes and elution in an EDTA/SDS-containing buffer. DNA fragments were treated with Proteinase K and recovered with a QIAquick PCR purification kit (Qiagen). Validation of the DRIP was performed by qPCR. Primer pairs used for DRIP analysis are described in Supplementary Information, Supplementary Table 1.

**DNA immunoprecipitation**. DNA immunoprecipitation (DIP) analysis was performed as described earlier[62] with minor adaptations. Briefly, homogenized cells in TE buffer were lysed overnight in 20 mM Tris–HCl, pH 8.0, 4 mM EDTA, 20 mM NaCl, 1% SDS at 37 °C with 20 μl Proteinase K. Genomic DNA was purified with Phenol: Chloroform, treated with RNAse A and sonicated with Diagenode Bioruptor to an average DNA length of 300-600 bp. Fragmented DNA was re-purified using Phenol: Chloroform extraction and 4 μg of re-purified DNA was was diluted in 450 μl TE buffer. DNA was denatured for 10 min at 95 °C and chilled on ice for additional 10 min. Fifty-one μl of 10x IP buffer (100 mM Na-Phosphate, pH 7.0, 1.4 M NaCl, 0.5% Triton X-100) and 4 μg of antibodies specific against 5mC (Abcam), dsDNA (Abcam) and IgG as a control (Santa Cruz) were added to the DNA-TE mix. After 2 h while overhead shaking at 4 °C, 40 μl of prewashed Dynabeads (0.1% BSA/PBS) were added for additional 2 h. Beads were washed 3 times in 1x IP buffer, followed by resuspension in 100 μl proteinase K digestion buffer (50 mM Tris–HCl, pH 8.0, 10 mM EDTA, 0.5% SDS). Precipitated DNA was removed from the beads by incubating with 100 μl of 50 mM Tris–HCl pH 8.0, 10 mM EDTA, 0.5% SDS and 5 μl Proteinase K (10 mg/ml stock) for 4 h at 37 °C. DNA was purified using the QIAquick PCR purification kit (Qiagen) and subjected to qPCR. For qPCR, the percentage of input was calculated after subtracting the IgG background.

**Comet assay**. Comet assay (Ancam) was performed as described by the manufacturer with minor modifications. MEF were treated with doxycycline for 6 h, harvested by trypzination and mixed with low-melting agarose. Mixture was immediately added onto microscopy slides. Lysis was performed in alkaline lysis solution (1.2 M NaCl, 100 mM Na$_2$EDTA, 0.1% sodium lauryl sarcosinate, 0.26 M NaOH, pH >13) overnight. Slides were washed and electrophoresed in 0.03 M NaOH, 2 mM Na$_2$EDTA (pH ~12.3) at 1 V/cm for 25 min. DNA was stained with DNA Vista Dye (Abcam) and images were taken with a confocal microscope. Intensities were measured using ImageJ. The tail length and the extended tail moment were calculated as measure for DNA damage.

**Trapping of HMGA2 and dot blot**. Dox-inducible MEF were treated with 1 mg/ml dox for 6 h followed by incubation in DPBS at pH 2 for 30 min at 37 °C. Cells were washed with DPBS containing 100 mM NaCNBH$_3$ (Sigma), or 100 mM NaCl as a control. Cells were harvested in hypotonic lysis buffer (50 mM Tris-HCl pH 8.0, 2 mM EDTA pH 8.0, 0.1% IGEPAL CA-630 (Sigma), 10% glycerol, 2 mM DTT and protease inhibitor cocktail) and nuclei were resuspended in 50 mM Tris-HCl pH 8.0, 5 mM EDTA pH 8.0, 1% SDS, 2 mM DTT and protease inhibitor cocktail. Nuclei-mix was briefly sonicated on ice and DNA was purified with Phenol: Chloroform. 200 ng of DNA was spotted onto a nitrocellulose membrane (GE Lifescience) for HMGA2 detection (Abcam; 1:1,000 dilution).

**GRO-seq, DRIPc-seq and R-loop analysis**. Raw reads from GRO-seq (MEF) and DRIP-seq (NIH/3T3 cells) experiments were downloaded from NCBI, trimmed with trimmomatic and high-quality reads were mapped using Bowtie2 to mouse genome mm10. GC skew values were obtained from the R-loop database (http://rloop.bii.a-star.edu.sg/gb2_database/mm10/bw/mm10.gc_skew.w200.s10.bw). Enrichment of R-loops, GC skew and nascent RNA (GRO-seq) in the Top15% candidate genes were analyzed using deeptools. Mean GC skew scores for the TSS plus 250 bp downstream region were obtained by using deeptools computeMatrix function with a bin size of 250. Genome browser snapshots from GRO-seq data were created with Homer using makeTagLibrary and makeUCSCfile. Available DRIPc plus-strand track file (http://rloop.bii.a-star.edu.sg/gb2_database/mm10/bw/mm9.To.mm10.GSM2104456_3T3_DRIPc_RNaseA.plus.bw) and GC skew track files were directly loaded into the genome browser.

**DNA nick assay and RNA DNA hybrid prediction**. Genomic DNA of *Hmga2* +/+, *Hmga2*−/− and doxycycline inducible MEF was extracted using the Gen-Elute DNA Miniprep kit (Sigma-Aldrich) according to the protocol provided by the manufacturer. Equal amount of total DNA was applied for Real-time PCR analysis. For detection of DNA nicks, primers for DNA nick assay were designed containing the consensus GT or CT sites specific for DNA nicking enzymes[38] (Supplementary Table 1). Nick primers were used with SYBR® Green on the Step One plus Real-time PCR system (Applied Biosystems) and normalized to the Ct values obtained within the surrounding ~300 bp DNA region amplified with the flanking primers (Supplementary Table 1). The % of nick DNA was represented as the ratio between: (Nick FWD + Flank RWD) / (Flank FWD + Flank RWD). To determine the directional association of the different ncRNAs associated to the nick DNA area close to the TSS on each target mRNA, we aligned the sequences of the associated ncRNAs using Global Alignment with free end gaps (Geneious 8.1.9, Biomatters Ltd., San Diego, CA). ncRNA:gDNA hybrids were predicted using the RNA hybrid-online server with parameter (MFE < -50 kcal/mol) and consensus alignment was calculated using T-coffee.

**DRIP-WB and DRIP-ChIP**. RNA/DNA hybrid IP was performed as described[63] with minor modifications. Non-crosslinked MEF were lysed in 10 mM HEPES pH 7.4, 10 mM KCl, 0.05% NP-40, 1 mM DTT, 25 mM NaF, 0.5 mM Na₃VO₄, 40 μg/ml phenylmethylsulfonyl fluoride and protease inhibitor. Nuclei were resuspended in RSB buffer (10 mM Tris-HCl pH 7.5, 200 mM NaCl, 2.5 mM MgCl₂) with 0.2% sodium deoxycholate, 0.1% SDS, 0.5% Sarkosyl and 0.5% Triton X-100 and sonicated 12 times with the Bioruptor. DNA was measured using Nanodrop and 50 μg of DNA for WB and 25 μg for ChIP was diluted 1:4 in RSB buffer supplemented with 0.5% Trion X-100. Three μg S9.6 antibody was added, and complexes were precipitated using preblocked protein A dynabeads (Invitrogen) for two hours while rotating. Beads were washed 4x with RSB supplemented with 0.5% Triton-X100 and eluted in 2x Laemmli buffer for WB or in TE buffer supplemented with 10 mM DTT for ChIP. Western blotting was performed using standard methods and antibodies specific for HIS-tag (Abcam; 1:1000 dilution), HA-tag (Santa Cruz; 1:1000 dilution), H3 (Abcam; 1:5000 dilution) were used. Immunoreactive proteins were visualized with the corresponding HRP-conjugated secondary antibodies (Jackson; 1:10,000 dilution) using the WesternBright ECL detection solutions (Biozym). Signals were detected and analyzed with Luminescent Image Analyzer (Las 4000, Fujifilm). The standard protocol described above was used for downstream ChIP.

**RNA sequencing and data analysis**. Total RNA of *Hmga2* +/+ and *Hmga2*−/− MEF treated with water or TGFB1 for 24 h was isolated using Trizol (Invitrogen). RNA was treated with DNase (DNase-Free DNase Set, Qiagen) and repurified using the miRNeasy micro plus Kit (Qiagen). Total RNA and library integrity were verified on LabChip Gx Touch 24 (Perkin Elmer). One μg of total RNA was used as input for SMARTer Stranded Total RNA Sample Prep Kit-HI Mammalian (Clontech). Sequencing was performed on the NextSeq500 instrument (Illumina) using v2 chemistry with 1x75bp single end setup. Raw reads were visualized by FastQC to determine the quality of the sequencing. Trimming was performed using trimmomatic with the following parameters LEADING:3 TRAILING:3 SLIDINGWINDOW:4:15 HEADCROP:5, MINLEN:15. High quality reads were mapped using with HISAT2 v2.1.0 with reads corresponding to the transcript with default parameters. RNA-seq reads were mapped to mouse genome mm10. After mapping, Tag libraries were obtained with MakeTaglibrary from HOMER (default setting). Samples were quantified by using analyzeRepeats.pl with the parameters (mm10 -count genes -strand + and –rpkm; reads per kilobase per millions mapped). UCSC known genes with a 1.5-fold change upon TGFB1 treatment in *Hmga2* +/+ MEF were classified as TGFB1 inducible and used for downstream analysis. To avoid division through zero, those reads with zero RPKM were set to 0.001.

**ChIP sequencing after Ctrl versus TGFB1 treatment and data analysis**. Precipitated DNA samples were purified by QIAquick PCR purification kit (Qiagen) and quantified by Qubit dsDNA HS Assay Kit (Thermo Fisher Scientific). Two ng of DNA was used as input for TruSeq ChIP Library Preparation Kit (Illumina) with following modifications. Instead of gel-based size selection before final PCR step, libraries were size selected by SPRI-bead based approach after final PCR with 18 cycles. In detail, samples were first cleaned up by 1x bead:DNA ratio to eliminate residuals from PCR reaction, followed by 2-sided-cleanup step with initially 0.6x bead:DNA ratio to exclude larger fragments. Supernatant was transferred to new tube and incubated with additional beads in 0.2x bead:DNA ratio for eliminating smaller fragments, like adapter and primer dimers. Bound DNA samples were washed with 80% ethanol, dried and resupended in TE buffer. Library integrity was verified with LabChip Gx Touch 24 (Perkin Elmer). Sequencing was performed on the NextSeq500 instrument (Illumina) using v2 chemistry with 1x75bp single end setup. Raw reads were visualized by FastQC to determine the quality of the sequencing. Trimming was performed using trimmomatic with the following parameters LEADING:3 TRAILING:3 SLIDINGWINDOW:4:15 HEADCROP:4, MINLEN:15. High quality reads were mapped by using with bowtie2[50] to mouse genome mm10. For downstream analysis, reads were scaled based on their read counts and normalized by subtracting reads of the corresponding inputs using deeptools.

**Overrepresentation analysis of top 15% genes and proteins enriched at TSS**. The top 15% candidate genes with UCSC ID were converted to refSeq using the UCSC genome Table browser. RefSeqs were submitted to DAVID Gene ID conversion tool (version 6.8) to obtain EntrezGene IDs. KEGG pathway enrichment analysis for EntrezGene IDs was performed using the enrichKEGG function from the clusterProfiler v3.0.4 package[64] with a *P* value cutoff of 0.05, a minimal size of genes annotated by Ontology term for testing of 10, a maximal size of genes annotated for testing of 500, a qvalue cutoff of 0.2. *P* Value was adjusted using the Benjamini & Hochberg method. Final plots were generated using the dotplot() function.

Overrepresented proteins identified by mass spectrometry in fractions 3 and 4 (1.65-fold enriched, *P* < 0.05) were identified using WEB-based GEne SeT AnaLysis Toolkit (http://www.webgestalt.org/) using KEGG and Reactome pathway as functional database.

**Identification and classification of ncRNAs associated with the top 15% candidate genes**. The distribution of the different ncRNA to the gene areas (introns, exons and 3′ UTR and TTS, transcription termination site +5 kb) and promoter (TSS, transcriptional start site, −5 kb) was analyzed using Bedtools v2.15 (intersect -wa -wb), crossing two bed files: the bed file containing the coordinates from the top 15% candidate genes ± 5 kb and the coordinates from the NONCODE database (http://www.noncode.org/datadownload/NONCODEv5_mm10.lncAndGene.bed.gz). In total, 52,976 annotated ncRNAs were found in 7,535 of the top 15% candidate genes (79%). For classification of the ncRNAs, duplicated ncRNAs due to isoforms in the top 15% candidate genes were considered only once which decreased the number of analyzed ncRNAs to 28,279. A bed-file containing the coordinates of the unique ncRNAs was analyzed using annotePeaks.pl for mm10 from Homer. Annotations in the 5′UTR or 3′UTR were counted as promoter or TTS enrichment, respectively. For orientation analysis, 7 ncRNAs were removed because of lacking information on transcription orientation. Fasta-sequences of divergent ncRNAs were submitted to MEME-ChIP (http://meme-suite.org/tools/meme-chip) and given strands were analyzed for motif enrichment using default parameters.

**Crossing of murine TGFB1 with human IPF data and GSEA**. IPF RNA-seq samples from GSE116086 (https://www.ncbi.nlm.nih.gov/geo/query/acc.cgi?acc=GSE116086) were remapped by the help of bowtie2 to human genome version hg38. Differential gene expression was analyzed using DEseq2 (default)[65]. Human gene name was converted to mouse (mgi_symbol) by the use of getLDS from biomaRt program. IFP-RNA-seq was crossed with the Hmga2-RNA-seq after TGFB1 treatment in *Hmga2* +/+ MEF. The log2 fold change (log2FC) both RNA-seq was used to perform a 2D kernel density plot by the help of the function kde2d from MASS package v7.3-51.4 with the number of grip points 50. Gene enrichment set analysis (GSEA) was obtained using fgsea (parameters minSize = 10, nperm = 1000) taken the "h.all.v7.0.symbols.gmt" as pathway database. PlotEnrichment was used to plot the two most enriched pathways from the Up-regulated genes from either IPF-RNA-seq or Hmga2-RNA-seq after TGFB1 treatment.

**Migration and proliferation assays**. Lung fibroblasts, to be assessed for cellular proliferation, were cultured either in 96-well or 48-well plates. Fibroblast proliferation was determined using colorimetric BrdU incorporation assay kit (Roche) according to manufacturer's instructions. Absorbance was measured at 370 nm with reference at 492 nm in a plate reader (TECAN). Depending on the experiment, proliferation of cells was plotted either as the difference of absorbance at 370 and 492 nm (A370 nm–A492 nm) or as a percentage of absorbance compared to control cells absorbance.

**Collagen assays**. Total collagen content was determined using the Sircol Collagen Assay kit (Biocolor, Belfast, Northern Ireland). Equal amounts of protein lysates from Ctrl and IPF human lung fibroblasts were added to 1 ml of Sircol dye reagent, followed by 30 min of mixing. After centrifugation at 10,000 × g for 10 min, the supernatant was carefully aspirated, and 1 ml of Alkali reagent was added. Samples and collagen standards were then read at 540 nm on a spectrophotometer (Bio-Rad). Collagen concentrations were calculated using a standard curve generated by using acid-soluble type 1 collagen.

**Hydroxyproline measurements**. Hydroxyproline levels in human lung fibroblasts were determined using the QuickZyme Hydroxyproline Assay kit (QuickZyme Biosciences). The cells and lung tissue were separately homogenized in 1 ml 6 N HCl with a Precellys tissue homogenizer (2 × 20 s, 3,800 g). The homogenate was then hydrolyzed at 90 °C for 24 h. After centrifugation at 13,000 *g* for 10 min, 100 μl from the supernatant was taken and diluted 1:2 with 4 N HCl. 35 μl of this working dilution was transferred to a 96-well plate. Likewise, a hydroxyproline standard (12.5–300 μM) was prepared in 4 N HCl and transferred to the microtiterplate. Following addition of 75 μl of a chloramine T-containing assay buffer, samples were oxidized for 20 min at room temperature. The detection reagent containing p-dimethylaminobenzaldehyde was prepared according to the manufacturer's instruction and 75 μl added to the wells. After incubation at 60 °C for 1 h, the absorbance was read at 570 nm with a microtiter plate reader (Infinite M200 Pro, Tecan) and the hydroxyproline concentration in the sample was calculated from the

standard curve and related to the employed amount of lung tissue. The hydroxyproline content in lung tissue is given as µg hydroxyproline per mg lung tissue.

**Experiments with human PCLS**. Control PCLS used for Supplementary Fig. 10d were prepared from tumor-free lung explants from patients who underwent lung resection for cancer. IPF PCLS used for Fig. 9i and Supplementary Figs. 10d–g were prepared from explanted lungs from IPF patients. Both types of tissue were obtained from the KRH Hospital Siloah-Oststadt-Heidehaus or the Hanover Medical School (both Hanover, Germany). Tissue was processed immediately within 1 day of resection as described before[66]. Briefly, human lung lobes were cannulated with a flexible catheter and the selected lung segments were inflated with warm (37 °C) low-melting agarose (1.5%) dissolved in DMEM Nutrient Mixture F-12 Ham supplemented with l-glutamine, 15 mM HEPES without phenol red, pH 7.2–7.4 (Sigma Aldrich), 100 U/ml penicillin, and 100 µg/mL streptomycin (both from Biochrom). After polymerization of the agarose solution on ice, tissue cores of a diameter of 8 mm were prepared using a sharp rotating metal tube. Subsequently, the cores were sliced into 300–350 µm thin slices in DMEM using a Krumdieck tissue slicer (Alabama Research and Development). PCLS were washed 3× for 30 min in DMEM and used for experiments. Viability of the tissue was assessed by an LDH Cytotoxicity Detection Kit (Roche) according to the manufacturer's instruction. For IPF resolution experiments, human IPF PCLS were treated with 50 or 100 µM FACTin for 72 h and the medium with FACTin replenished every 24 h.

**Immunofluorescence staining in PCLS**. PCLS from IPF patients were fixed with acetone/methanol (Roth) 50:50 by volume for 20 min, blocked for 1 h with 5% bovine serum albumin (w/v, Sigma) in 1x PBS, pH 7.4. Cells were then incubated with primary antibody overnight at 4 °C. After incubation with a secondary antibody for 1 h, nuclei were DAPI stained and PCLS were examined with a confocal microscope (Zeiss). Antibodies used were specific for DNA-RNA hybrid (S9.6, Kerafast; 1:500 dilution), COL1A1 (Sigma; 1:500 dilution), ACTA2 (Sigma; 1:500 dilution), FN1 (Millipore; 1:500 dilution), VIM (Cell Signaling; 1:500 dilution), pH2A.X (Millipore; 1:500 dilution) and HMGA2 (Santa Cruz; 1:200 dilution). Alexa 488, Alexa555 or Alexa 594 tagged secondary antibodies (Invitrogen) were used. DAPI (Sigma Aldrich) used as nuclear dye.

**Statistics and reproducibility**. The source data for all the plots presented in the article, including the values for statistical significance and the implemented statistical tests, are provided in Source Data file 01. Further details of statistical analysis in different experiments are included in the Figures and Figure legends. Briefly, protein enrichment on chromatin and expression analysis of samples were analyzed by next generation sequencing in duplicates of one experiment. Three independent experiments of the mass spectrometry-based proteomic approach were performed. For the rest of the experiments presented here, samples were analyzed at least in triplicates and experiments were performed three times. Statistical analysis was performed using GraphPad Prism 5 and Microsoft Excel. Data in bar plots are represented as mean ± standard error (mean ± s.e.m.). t-Tests were used to determine the levels of difference between the groups and $P$-values for significance. $P$-values after one- or two-tailed t-test, *$P \leq 0.05$; **$P \leq 0.01$ and ***$P \leq 0.001$. In the Figs. 1b, 3b, 4f, 8b, 8c and Supplementary Figure 9b, $P$-values were determined using Wilcoxon–Mann–Whitney test. In the 2D Kernel Density plot presented in Fig. 9a the statistical significance was calculated using DESeq2's integrated Wald test.

**Reporting summary**. Further information on research design is available in the Nature Research Reporting Summary linked to this article.

## Data availability

The data that support this study are available from the corresponding author upon reasonable request. In addition, sequencing data of ChIP, RNA and MNase have been deposited in NCBI's Gene Expression Omnibus[67] and is accessible through GEO Series with accession number GSE141272. The mass spectrometry-based interactome data have been deposited into the PRIDE archive and assigned to the project accession PXD016586. In addition, we retrieved and used a number of publicly available datasets to aid analysis of our data:

pH2A.X and HMGA2 ChIP-Seq data: GSE63861[10]
DRIP-Seq in NIH/3T3 cells data: GSE70189[40]
Nuclear RNA-seq in Ctrl and human IPF hLF: GEO: GSE116086[44]
GRO-seq in Mouse embryonic fibroblasts: GEO: GSE76303[68]
DRIPc plus strand files: [http://rloop.bii.a-star.edu.sg/gb2_database/mm10/bw/mm9.To.mm10.GSM2104456_3T3_DRIPc_RNaseA.plus.bw]
GCskew [http://rloop.bii.a-star.edu.sg/gb2_database/mm10/bw/mm10.gc_skew.w200.s10.bw]
The source data are provided with this paper.

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

## Acknowledgements

We thank Roswitha Bender for technical support. Guillermo Barreto was funded by the "Université Paris-Est Créteil" (UPEC, Créteil, France), the Max-Planck-Society (MPG, Munich, Germany), the "Deutsche Forschungsgemeinschaft" (DFG, Bonn, Germany) (BA 4036/4-1), the "Centre National de la Recherche Scientifique" (CNRS, France), "Délégation Centre-Est" (CNRS-DR6) and the "Lorraine Université" (LU, France) through the initiative "Lorraine Université d'Excellence" (LUE) and the dispositif "Future Leader". Karla Rubio was funded by the "Consejo de Ciencia y Tecnología del Estado de Puebla" (CONCYTEP, Puebla, Mexico) through the initiative International Laboratory EPIGEN. Indrabahadur Singh is funded by the DFG (Bonn, Germany) through Emmy Noether program (SI 2620/1-1). Rafael Castillo received a doctoral fellowship from CONACyT (2019-000003-01EXTF-00156, Mexico). Jürgen Bernhagen acknowledges support from DFG under Germany's Excellence Strategy within the framework of the Munich Cluster for Systems Neurology (EXC 2145 SyNergy – ID 390857198) and within the LMU-EXC strategic cooperations program LMU/Singapore. Malgorzata Wygrecka acknowledges fundings from the DFG (WY119/1-3) and the German Center for Lung Research. Work in the lab of Thomas Braun is supported by the Deutsche Forschungsgemeinschaft, Excellence Cluster Cardio-Pulmonary Institute (CPI), Transregional Collaborative Research Center TRR81, TP A02, SFB1213 TP B02, TRR 267 TP A05 and the German Center for Cardiovascular Research. Biomaterials wereprovided by the European IPF Registry and Biobank (eurIPFreg).

## Author contributions

S.D., K.R., I.S., S.G., J.G., J.C., R.C.N., P.B., A.M., M.B.H. and G.B. designed and performed the experiments. H.C.F., J.B., C.M.C., S.B., A.G., K.T.P., S.K., G.D., M.W., T.B. and D.P.G. were involved in study design. G.B., S.D., K.R., I.S., S.G., J.G. and J.C. designed the study, analyzed the data. G.B., S.D., K.R. and I.S. wrote the manuscript. All authors discussed the results and commented on the manuscript.

## Funding

## Competing interests

G.B. was scientific advisor for a company in USA. The remaining authors declare that they have no competing interests.

## Additional information

[1]Univ Paris Est Creteil, Glycobiology, Cell Growth and Tissue Repair Research Unit (Gly-CRRET), Brain and Lung Epigenetics (BLUE), Creteil, France. [2]Lung Cancer Epigenetic, Max-Planck-Institute for Heart and Lung Research, Bad Nauheim, Germany. [3]Human Biology Division, Fred Hutchinson Cancer Research Center, Seattle, WA, USA. [4]Laboratoire IMoPA, UMR 7365 CNRS-Université de Lorraine, Biopôle de l'Université de Lorraine, Campus Biologie-Santé, Faculté de Médecine, Vandœuvre-lès-Nancy Cedex, France. [5]International Laboratory EPIGEN, Universidad de la Salud del Estado de Puebla, Puebla, Mexico. [6]Emmy Noether Research Group Epigenetic Machineries and Cancer, Division of Chronic Inflammation and Cancer, German Cancer Research Center (DKFZ), Heidelberg, Germany. [7]ECCPS Bioinformatics and Deep Sequencing Platform, Bad Nauheim, Germany. [8]Department of Cardiac Development, Max-Planck-Institute for Heart and Lung Research, Bad Nauheim, Germany. [9]Biomolecular Mass Spectrometry, Max-Planck-Institute for Heart and Lung Research, Bad Nauheim, Germany. [10]Anatomy and Developmental Biology, CBTM, Mannheim, Germany. [11]European Center for Angioscience (ECAS), Medical Faculty Mannheim, Heidelberg University, Mannheim, Germany. [12]Univ Paris Est Creteil, Glycobiology, Cell Growth and Tissue Repair Research Unit (Gly-CRRET), F-, Creteil, France. [13]Pharmaceutical Technology and Biopharmaceutics, Department of Pharmacy, Ludwig-Maximilians-University (LMU) Munich, Munich, Germany. [14]Institute for Pathology, Hannover Medical School, Hanover, Germany. [15]Biomedical Research in Endstage and Obstructive Lung Disease Hannover (BREATH) Research Network, Hanover, Germany. [16]Faculty of Medicine, Department of Biochemistry, Justus Liebig University, Giessen, Germany. [17]National Heart Research Institute, National Heart Centre Singapore, Singapore, Singapore. [18]Tecnologico de Monterrey, Centro de Biotecnologia-FEMSA, Monterrey, Mexico. [19]Cardiovascular and Metabolic Disorders Program, Duke-National University of Singapore Medical School, Singapore, Singapore. [20]Chair of Vascular Biology, Institute for Stroke and Dementia Research (ISD), LMU University Hospital, Ludwig-Maximilians-University (LMU) Munich, Munich, Germany. [21]Munich Cluster for Systems Neurology (SyNergy), Munich, Germany. [22]Cardio-Pulmonary Institute, Giessen, Germany. [23]International Collaborative Center on Growth Factor Research, School of Pharmaceutical Sciences, Wenzhou Medical University and Institute of Life Sciences, Wenzhou University, Wenzhou Zhejiang, China. [24]Member of the Universities of Giessen and Marburg Lung Center (UGMLC), Giessen, Germany. [25]German Center of Lung Research (Deutsches Zentrum für Lungenforschung, DZL), Giessen, Germany. [26]Pulmonary and Critical Care Medicine, Department of Internal Medicine, Justus Liebig University, Giessen, Germany. [27]Agaplesion Lung Clinic Waldhof Elgershausen, Greifenstein, Germany. [28]Center for Infection and Genomics of the Lung (CIGL), Universities Giessen and Marburg Lung Center, Giessen, Germany. ✉email: guillermo.barreto@u-pec.fr

