## [Peer Review File · Nature Communications]

REVIEWER COMMENTS

Reviewer #1 (Remarks to the Author):

This manuscript is focused on analysis of gamma-H2AX nucleosome positioning to establish that it precedes active DNA demethylation and subsequent transcription initiation. Overall the manuscript is very clear with the order of events established and complementary inhibitor studies that clearly support the proposed order (Fig 5B). The experiments with FACTin using IPF cells clearly show a reduction in fibrosis markers and use of this inhibitor seems to be a promising therapeutic for this condition. With additional context, the proposed transcription mechanism is truly revolutionary – that a DNA nick is created to allow exchange of H2A for H2A.X by FACT has not been examined before (at least not to my knowledge).

Major Concerns:

Lack of quantitative narrative throughout the results section. Most of the narrative is summary statements rather than writing about the values they obtained. The quantitative values obtained and their relevance to the narrative summaries should be established in the text when they are key to the conclusions. This is a data-rich paper and it would be helpful if the authors would guide the reader through the quantitative data to fully synthesize the data shown.

All three gene examples can be somehow linked to the study candidates (from previous studies or by their top 15% parameter). Perhaps a 4th gene example that has no previous data linking it to anything in this study, and also is not found in the “top 15%” would help to serve as a control. Tuba1A was used to normalize qRT-PCR data, this could be an option.

Treatment of all three gene examples as the same when the data doesn't always support it. Differences between observations for each gene should be noted.

Lack of consistency of y-axis values within the same experiment. In some cases, there is a 10 fold difference in values, this should be altered or an explanation should be given on why this is required for clarity in response to the review.

General lack of context for the newly proposed transcription initiation mechanism that involves DNA nicks in the field of transcription initiation (both in the introduction and particularly the discussion sections). There is an opportunity here to explore the potential mechanisms in the discussion with a forward-looking perspective.

There are many experiments, which is appreciated, however the narrative is a bit overwhelming and difficult to follow. The authors should be careful to expand the narrative so that non-experts in either the field or with these types of data can follow along (and also as discussed above for the quantitative data).

SAINT (Significance Analysis of INTERactome) analysis would be a nice complement to NES panel in figure 7B. This software is freely available through the Crapome.org server along with descriptions on how to format the data and implement the analysis.

Overexpression of HGMA2 with a C-terminal MYC and HIS is utilized for a Co-IP of nuclear protein extracts to characterize interactions between FACT and HMG2A. Please address how the overexpression of HGMA2 differs from the endogenous HGMA2 levels.

Minor Concerns:

Number of reps (n=) is not always clearly stated and should be included in at least the figure legend if not also in the y-axis labels or within specific panels when appropriate.

Figure 4 legend has no explanation for panel E. It is unclear from the legend or the results section how the DNA nicks were detected. It is also very difficult to see the DNA/RNA hybrids and the explanation for how they form (lines 320-324) is unclear.

Figure 7H has no statistical analysis shown for the H-proline data

The data from Figure 6D-G don't line up well with the data from 7D-F where they are measuring the same parameters and the only difference is the type of cells used as inputs (mouse for Fig 6, human for 7). These differences should be noted in the results narrative or the discussion.

In Figure 5H, the legend describes the inclusion of a hmga2 specific short hairpin DNA but why this was included in the transfection is unclear from the methods or results narrative.

Colors for Figure 1 should be explained in the legend, the H3 data in that figure should have additional narrative in the results for clarity.

The supplemental figure S5 should be moved to the main manuscript

Reviewer #2 (Remarks to the Author):

Within the manuscript "Positioning of nucleosomes containing γ -H2AX precedes active DNA demethylation and transcription initiation" Stephanie Dobersch and colleagues report a more defined mechanism of the high-mobility group protein HMGA2 role in transcription initiation. Specifically, they demonstrate that HMGA2 cuts the DNA within R-loops at transcription start sites, which in turn allows the complex FACT to deposit H2A.X and ATM to phosphorylate this histone, leading to DNA repair and DNA demethylation mediated by GADD45A and TET1. Furthermore, having previously demonstrated a role for HMGA2 in TGFB1 signaling, the authors confirmed their proposed mechanism in this context.

Finally, they evaluated the clinical potential if this mechanism in the lung disease IPF (idiopathic pulmonary fibrosis), in which TGF β signaling plays a key role. They find that inhibiting the FACT complex activity in this disease model decreases fibrosis hallmarks, revealing new potential therapeutic strategies for IPF. The authors report here an impressive body of work that not only improves our understanding of the role of chromatin regulation in transcriptional control but also connects its (mis)regulation to diseases, bringing new therapeutic insights into a non-curable disease. Please find below specific points that I would suggest to be addressed.

Major comments

1- Figure 3A and lane 195: the authors argue that pH2A.X mainly sedimented in fraction 4, on the contrary to non-modified histones that sedimented in fractions 1 to 4. By looking closely at the reported western blots it looks like a longer exposure would show pH2A.X sedimentation identical to the non-modified histones: enriched in fraction 4 but also present in fractions 1, 2, 3, and 5. This can be seen in both Hmga2 $+/+$ and $-/-$ blots. A quantification of several western blots and longer exposures of pH2A.X would be necessary to decipher this point.

2- Figure 3A and lane 198: the authors conclude to a reduction of pH2A.X levels in Hmga2 $-/-$ conditions. How do they come to that conclusion? Looking at the input pH2A.X levels look reduced, but not looking at the fractions: $-/-$ fractions 4 and 5 together could equal fraction 4 in $+/+$, in which case a redistribution could be another conclusion. Here also a quantification of several blots would be necessary, as well as another method such as immunofluorescence to assess total levels.

3- Figure 3A and lane 201: related the previous points, the authors decide to focus on fractions 3 and 4 because pH2A.X is present in 4 and absent in 3. We can clearly see pH2A.X in fraction 3 in the $-/-$ condition, it could therefore be that pH2A.X is distributed differently among fractions based on the different conditions, which could be the reason to focus on these fractions. Once again additional blots, longer exposures and proper quantifications would be necessary.

4- Figure 3D and lane 210: the authors conclude of the reduction of SUPT16 and SSRP1 in fraction 4 but not in fraction 3. This is not what the western blot reports: both proteins seems reduced to the same extend in both fractions. As for the other points, additional blots and quantifications would be helpful.

Minor comments

1- Citation of Figure S4F is missing lane 296.

Reviewer #3 (Remarks to the Author):

Transcription and DNA repair were initially considered separate processes occurring at the chromatin. During recent years, however, these processes have been shown to share many protein components, and demonstrated to be intertwined (reviewed e.g. in Fong, Cattoglio and Tijan, 2013, Mol. Cell). Here, Dobersch et al., shed light on the interacting mechanisms of generating DNA nicks, recruiting DNA repair machinery and transcriptional activation of methylated DNA. According to the proposed model, high mobility group protein HMGA2 induces DNA nicks at the TSS of methylated DNA. Subsequently, histone chaperone FACT positions histone variant H2A.X into adjacent nucleosomes, and ATM phosphorylates

the H2A.X at the Ser-139. pH2A.X, which hallmarks double-stranded DNA breaks, then attracts DNA-repair machinery to repair the nicks and demethylate the DNA. As a result, the unmethylated DNA can support transcriptional activation of the gene.

The Barreto group has previously shown HMGA2-mediated transcriptional activation to require phosphorylation of H2A.X (Singh et al., 2015; <https://www.nature.com/articles/cr201567>). Moreover, DNA nicks have been linked to demethylation of the DNA and transcriptional activation (reviewed e.g. in Fong, Cattoglio and Tijan, 2013, Mol. Cell). In this study, the authors extend the previous findings by mechanistically dissecting the HMGA2-FACT-ATM-pH2A.X axis at methylated DNA. The study utilizes mouse embryonic fibroblasts (MEFs) derived from wildtype and HMGA2 KO mice, and assays H2A.X positioning, and Pol II levels at the chromatin. Moreover, protein constituents of selected, MNase-digested chromatin fractions are identified using mass spectrometry. To dissect the order of events from HMGA2 binding to demethylation of DNA, the study uses WT and HMGA2 KO mice, inducible HMGA2 mutant with unfunctional AT-hook, chemical inhibition of FACT and ATM, and downregulation of GADD45. The study is in general well conducted and communicated, and the results are up-to date and relevant. However, certain aspects should be clarified before publication.

Major concerns:

- 1) The general claim of the subtitle “Position of the first nucleosome containing pH2A.X determines the basal transcription” is inadequately backed up by the data. In the study, genes are first ranked based on the pH2A.X levels at the promoter, and the top 15% of genes clustered into three groups using positional information of the pH2A.X (ChIP-seq data). Then, HMGA2 and Pol II with Ser-5 phosphorylated CTD (pPol II) are shown to co-localize with the pH2A.X at these selected genes (Fig. 2E). Finally, transcriptional activity of genes in the three clusters are deduced from mRNA-seq data. Although there is a statistical significance between the mRNA expressions of the distinct groups (Figure 2F), there is a lot of variation within genes, and no causal support for pH2A.X position to determine the transcriptional activity. Please note that the top 15% of genes with the highest pH2A.X level appear in general to be lowly transcribed (based on pPol II ChIP-seq and GRO-seq data, discussed further in points 2 and 3 below). Rather, the data supports a conclusion where pPol II positioning at the TSS correlates with the positioning of the pH2A.X and HMG2A. Moreover, the positions of the nucleosomes (including +1) are not shown. Please, refine the subtitle to match the results and include data (e.g. MNase-seq or H3 ChIP-seq) to report the nucleosomal positions.
- 2) The quality of the pPol II ChIP-seq data should be adequately demonstrated. Please note that Ser5 phosphorylated Pol II has been shown to extend beyond the promoter-proximal region (e.g. by Kim...Bentley, 2010, cited in this manuscript, and Zaborowska...Murphy, 2016). The spiky pPol II browser shot images (Figure 2C) could be due to the low level of transcription of these example genes (displaying methylated DNA at the promoter). Showing a highly transcribed gene as an example and giving the range of transcriptional activity for the top 15% genes would be helpful in interpreting the results.
- 3) H3 profile is used for detecting +1 nucleosomes in example genes in Figure 1D. Based on these profiles, there seem to be re-positioning of nucleosomes in addition to the reduction in H2A.X levels in HMGA2-/- MEFs. Do the nucleosomal positions show a general change in the absence of HMGA2?

4) The top 15% genes based on pH2A.X levels are shown to display a remarkable loss of pPol II and pH2A.X in the absence of HMGA2 (Figure 2E). Is this loss specific to the top 15% genes or a more general phenomenon across the genome? Please discuss the phenotype of the HMGA2 KO mice and refer to the study generating the mouse.

5) Lines 317-320 state nascent non-coding RNAs to be found upstream of 46% of the top candidates. Are these ncRNAs divergent transcripts that are generated by mammalian promoters? See e.g. Jonkers et al., eLife, 2014 that shows 95% of genes in mouse to have divergent transcription. Nevertheless, the nature of these ncRNAs should be disclosed.

6) The section on the R-loops matching the gene's promoter proximal region requires clarification: Are the ncRNAs aligned to the coding regions of the upstream gene? What is the alignment rate/score? Please note that the DNA-RNA structures (right panels of Fig 4F) are very difficult to see.

Minor comments:

1) The section: "HMGA2-FACT interaction and -lyase activity are required for pH2A.X deposition and solving of R-loops" is very long. Please consider dividing the HMGA2-FACT interaction and R-loops into their own subtitled sections.

2) Please check the sentence in lines 197-199, it seems to miss something and the logic is hard to follow.

3) In Figure 3D, DNA is visible only in fractions 3-5. Why isn't DNA detected in the other fractions, such as 1 and 2 that harbor abundant levels of Pol II?

4) One of the strengths of this study is the disruption of distinct molecules in the HMGA2-FACT-AMT-pH2A.X pathway. However, many inhibitory molecules have side effects and might not be specific to a certain function of the target molecule. Please discuss the mechanism of inhibition and the known selectivity of the FACTin and ATMin. E.g. is it known how CBLC000 trifluoroacetate blocks FACT into a chromatin-bound state and which functions of FACT does the compound inhibit? What is the functional mechanism that KU-55933 exhibits on ATM?

5) Certain sections of the methods should be clarified. E.g. clustering the top 15% genes is vaguely explained (see lines 894-897). Why were only genes with RPKMs less than 10 included in the analyses? This further confuses the claim on pH2A.X determining transcriptional activity (see major concern 1). Likewise, the analyses of downloaded GRO-seq and DRIP-seq data, as well as the R-loop prediction are inadequately described.

6) Please consider structuring the discussion with subtitles.

Point-by-point response to the Reviewers – Manuscript with the number NCOMMS-20-12849 and the title “Positioning of nucleosomes containing γ -H2AX precedes active DNA demethylation and transcription initiation” submitted as Research Article to Nature Communications by Dobersch et al., 2020.

REVIEWER COMMENTS

Reviewer #1 (Remarks to the Author):

This manuscript is focused on analysis of gamma-H2AX nucleosome positioning to establish that it precedes active DNA demethylation and subsequent transcription initiation. Overall the manuscript is very clear with the order of events established and complementary inhibitor studies that clearly support the proposed order (Fig 5B). The experiments with FACTin using IPF cells clearly show a reduction in fibrosis markers and use of this inhibitor seems to be a promising therapeutic for this condition. With additional context, the proposed transcription mechanism is truly revolutionary – that a DNA nick is created to allow exchange of H2A for H2A.X by FACT has not been examined before (at least not to my knowledge).

We would like to thank Reviewer 1 for the positive comments related to the novelty and translational potential of our work and for the time and efforts implemented during the peer review of our manuscript.

To facilitate the recognition of new added results during the revision of our manuscript, we have submitted two pdf files for Reviewers only, in which we highlighted in blue the corresponding text in the manuscript, as well as the figure panels in the main and supplementary figures (please see pdf file containing main text, main and supplementary figures for Reviewers only). Following the constructive suggestions from the Reviewers led us to a substantially improved manuscript.

Major Concerns:

1. Lack of quantitative narrative throughout the results section. Most of the narrative is summary statements rather than writing about the values they obtained. The quantitative values obtained and their relevance to the narrative summaries should be established in the text when they are key to the conclusions. This is a data-rich paper and it would be helpful if the authors would guide the reader through the quantitative data to fully synthesize the data shown.

We completely agree with this point of Reviewer 1. We followed this constructive suggestion from Reviewer 1 and introduced in several places of the revised version of the manuscript the quantitative values that were obtained, especially when they are key to the conclusions. To facilitate the recognition of the changes done to the manuscript, an additional pdf file of the manuscript has been submitted, with the changes highlighted in blue.

2. All three gene examples can be somehow linked to the study candidates (from previous studies or by their top 15% parameter). Perhaps a 4th gene example that has no previous data linking it to anything in this study, and also is not found in the “top 15%” would help to serve as a control. *Tuba1A* was used to normalize qRT-PCR data, this could be an option.

We followed this suggestion from Reviewer1. However, instead of *Tuba1A*, we selected *Rptor* as negative control since *Rptor* is outside the top 15% candidate genes, having a similar expression level as the selected *Hmga2* target genes (please see comments from Reviewer 3 related to basal expression levels of *Hmga2* target genes and the newly added Supplementary Fig. 2). Keeping these two comments in mind, we believe *Rptor* will be more adequate as negative control than *Tuba1A*. We hope that Reviewer 1 will share our opinion.

3. Treatment of all three gene examples as the same when the data doesn't always support it. Differences between observations for each gene should be noted.

We recognize and share the attention to detail and accuracy from Reviewer 1. Following this line of ideas, we have noted the differences between the results obtained from the selected Hmga2 target genes in the manuscript, whenever they are key to the conclusions. Further, we beg Reviewer 1 for his/her understanding, if in some cases differences between the selected target genes have not been noted, to avoid any incomprehensiveness of the text.

4. General lack of context for the newly proposed transcription initiation mechanism that involves DNA nicks in the field of transcription initiation (both in the introduction and particularly the discussion sections). There is an opportunity here to explore the potential mechanisms in the discussion with a forward-looking perspective.

We appreciate this good observation and corresponding constructive suggestions from Reviewer 1. We have included in the Introduction (lines 101-104), the Results (lines 417-422) and in the Discussion (lines 592-596) new text and references following the suggestions from Reviewer 1.

5. Lack of consistency of y-axis values within the same experiment. In some cases, there is a 10 fold difference in values, this should be altered or an explanation should be given on why this is required for clarity in response to the review.

Please, allow me to explain, why we do not interpret as “lack of consistency” the differences in the values obtained for the y-axis. Even though the same method (e.g. ChIP) might be used for the analysis of the same gene (e.g. Gata6), the values obtained might differ because of different aspects, including:

- the use of different source of material (e.g. chromatin from cell lines, primary cells, precision-cut lung slices, murine or human material, etc.).
- the use of different antibodies with different affinity of their specific epitopes. The production of some of the antibodies has been discontinued.
- the implementation of different equipment (sonicator, qPCR-machine, etc) in different laboratories by different operators.
- Last but not least, differences due to biological replicates.

All these aspects might have an additive effect resulting in differences from experiment to experiment. However, even with these differences in the absolute values, the interpretation of the results relative to the controls and to the other conditions in the experiment support and strengthen our hypothesis and our conclusions. We hope that we were able to clarify this specific point.

6. There are many experiments, which is appreciated, however the narrative is a bit overwhelming and difficult to follow. The authors should be careful to expand the narrative so that non-experts in either the field or with these types of data can follow along (and also as discussed above for the quantitative data).

We completely agree with this point from Reviewer 1. In the revised version of our manuscript, we expanded the narrative and improved the description and interpretation of results, making the text more comprehensive for a wider audience. In addition, we would like to refer to the answer from point 1 for further explanations, since both points (1. and 6.) are related to each other.

7. Overexpression of HGMA2 with a C-terminal MYC and HIS is utilized for a Co-IP of nuclear protein extracts to characterize interactions between FACT and HMG2A. Please address how the overexpression of HGMA2 differs from the endogenous HGMA2 levels.

We have addressed this point in Supplementary Fig. 5b. We induced the expression for 24h by doxycycline treatment and compared the expression levels to *Hmga2*^{+/+} and *Hmga2*^{-/-} MEF by using an HMGA2 specific antibody. We found that exogenous expression is higher in transgenic cell lines but is still in a comparable range to *Hmga2*^{+/+} MEF.

8. SAINT (Significance Analysis of INTeractome) analysis would be a nice complement to NES panel in figure 7B. This software is freely available through the Crapome.org server along with descriptions on how to format the data and implement the analysis.

We thank Reviewer 1 for this specific comment. We consulted four of the co-authors of the manuscript, who are specialist on next-generation sequencing (NGS) and proteomic approaches. Since SAINT analysis was originally designed for proteomic data analysis and the data presented in the previous Fig. 7b (now Fig. 9b) are RNA-seq data, all consulted co-authors were of the opinion that a further analysis of the RNA-seq data from Fig. 9b using SAINT will not necessarily contribute to a deeper understanding of the data, or identification of new perspectives, and could rather lead to a potential over interpretation of the data. For these reasons, we decided not to include this type of analysis into the manuscript. We beg Reviewer 1 for her/his understanding.

Minor Concerns:

9. Number of reps (n=) is not always clearly stated and should be included in at least the figure legend if not also in the y-axis labels or within specific panels when appropriate.

Following the suggestions from Reviewer 1, in the revised version of our manuscript the number of reps is stated in the Fig. legends, as well as in the statistical summary containing all relevant information from the plots presented in the manuscript, which is provided as Source Data file. In each Fig. legend we refer the reader to the Source Data file.

10. Figure 4 legend has no explanation for panel E. It is unclear from the legend or the results section how the DNA nicks were detected. It is also very difficult to see the DNA/RNA hybrids and the explanation for how they form (lines 320-324) is unclear.

We thank Reviewer 1 for these observations, which led us to improve the structure of our manuscript facilitating the understanding. We would like to summarize these changes in the following points:

- The previous Fig. 4 was divided into Fig. 5 and 6. This allows us to present the DNA/RNA hybrids in the new Fig. 6c with higher magnification and including additional information, such as consensus analysis.
- The previous Fig. 4e is now Fig. 6b. The legend of the new Fig. 6b has been included into the new version of the manuscript.
- Moreover, to facilitate the understanding of the results presented in the new Figs. 5 and 6, we generated a new Supplementary Fig. 7, which is described in the lines 336-359.

11. Figure 7H has no statistical analysis shown for the H-proline data

We thank Reviewer 1 for this sharp observation. The data are now in the new Fig. 9h and the statistical analysis has been included. The Source Data file containing the statistical summary of the data presented in the manuscript has been accordingly updated.

12. The data from Figure 6D-G don't line up well with the data from 7D-F where they are measuring the same parameters and the only difference is the type of cells used as inputs (mouse for Fig 6, human for 7). These differences should be noted in the results narrative or the discussion.

We disagree with this comment from Reviewer 1 as elaborated in the following points:

- Although there are slight differences that are expected by the different experimental systems implemented for the results presented in the previous Figs. 6d-g (new Figs. 8d-g) and previous Figs. 7d-f (new Figs. 9d-f), it is relevant to mention that the results presented in both figures correlate with each other and, more importantly, support our interpretation and strengthen our model.
- The difference between the experimental models used in both figures are not limited to the species of origin, as the Reviewer 1 simplifies in the comment. In the new Fig. 8d-g wild type or *Hmga2*^{-/-} mouse embryonic fibroblasts (MEF) were treated with TGFB1, FACT inhibitor or Gadd45a-specific siRNA as

indicated. On the other hand, in the new Fig. 9d-f, precision-cut lung slices (PCLS) from control donor or IPF patients were treated with FACT inhibitor.

- In other words, the experimental system used for the new Fig. 8d-g is fine tuned for obtaining mechanistic insides, whereas the experimental system used for the new Fig. 9d-f is more complex due to 3-dimensional structure and the participation of different cell types of the human lung, thereby being closer to the pathophysiology of idiopathic pulmonary fibrosis and more suitable to determine the translational potential of the findings obtained using MEF.

We beg for the understanding of Reviewer 1.

13. In Figure 5H, the legend describes the inclusion of a hmga2 specific short hairpin DNA but why this was included in the transfection is unclear from the methods or results narrative.

We thank Reviewer 1 for attracting our attention to the previous Fig. 5h (new Fig. 7h). We have performed this experiment in MLE-12 cells due to the low transfection efficiency of expression constructs in MEF. We performed first the knockdown of HMGA2 by shRNA constructs and afterwards overexpression of WT or lyase-mutant HMGA2. Please see the newly added text in lines 444 to 447.

14. Colors for Figure 1 should be explained in the legend, the H3 data in that figure should have additional narrative in the results for clarity.

Following the suggestion from Reviewer 1, the colors are now explained in the legend of Fig. 1. In addition, we have written a short explanation about H3 ChIP-seq in line 152 of the Results section.

15. The supplemental figure S5 should be moved to the main manuscript.

We appreciate the interest of Reviewer 1 on this specific part of the manuscript. However, we decided not to include the previous Supplementary Fig. 5 (new Supplementary Fig. 8) as main Fig. of the revised manuscript, because we believe that it will interfere with the logical order of argumentation through the text. Even though Gadd45a plays an important role in the proposed mechanism, we do not want to attract the attention of the reader to this specific protein, since this manuscript is focused on other proteins of the proposed mechanism. We beg Reviewer 1 for her/his understanding.

We thank once again Reviewer 1 for the time and the efforts implemented for the peer review of our manuscript. The comments from Reviewer 1 are constructive and guided us to significantly improve the quality of our manuscript.

Reviewer #2 (Remarks to the Author):

Within the manuscript “Positioning of nucleosomes containing γ -H2AX precedes active DNA demethylation and transcription initiation” Stephanie Dobersch and colleagues report a more defined mechanism of the high-mobility group protein HMGA2 role in transcription initiation. Specifically, they demonstrate that HMGA2 cuts the DNA within R-loops at transcription start sites, which in turn allows the complex FACT to deposit H2A.X and ATM to phosphorylate this histone, leading to DNA repair and DNA demethylation mediated by GADD45A and TET1. Furthermore, having previously demonstrated a role for HMGA2 in TGFB1 signaling, the authors confirmed their proposed mechanism in this context. Finally, they evaluated the clinical potential of this mechanism in the lung disease IPF (idiopathic pulmonary fibrosis), in which TGFB signaling plays a key role. They find that inhibiting the FACT complex activity in this disease model decreases fibrosis hallmarks, revealing new potential therapeutic strategies for IPF. The authors report here an impressive body of work that not only improves our understanding of the role of chromatin regulation in transcriptional control but also connects its (mis)regulation to diseases, bringing new therapeutic insights into a non-curable disease. Please find below specific points that I would suggest to be addressed.

We thank Reviewer 2 for her/his positive opinion on our work and for the flattering and motivating comments. We also appreciate the time and the efforts implemented by Reviewer 2 during the peer review of our manuscript.

To facilitate the recognition of new added results during the revision of our manuscript, we have submitted two pdf files for Reviewers only, in which we highlighted in blue the corresponding text in the manuscript, as well as the figure panels in the main and supplementary figures (please see pdf file containing main text, main and supplementary figures for Reviewers only). Following the constructive suggestions from the Reviewers led us to a substantially improved manuscript.

Major comments:

1- Figure 3A and lane 195: the authors argue that pH2A.X mainly sedimented in fraction 4, on the contrary to non-modified histones that sedimented in fractions 1 to 4. By looking closely at the reported western blots it looks like a longer exposure would show pH2A.X sedimentation identical to the non-modified histones: enriched in fraction 4 but also present in fractions 1, 2, 3, and 5. This can be seen in both Hmga2 +/+ and -/- blots. A quantification of several western blots and longer exposures of pH2A.X would be necessary to decipher this point.

We agree with the observations from Reviewer 2. Thus, following the suggestions from Reviewer 2, we have changed the description of the previous Fig. 3a (new Fig. 4a) in lines 204-211 in the Results section of the revised manuscript as follows:

“Interestingly, the histone variant H2A.X and its post-translationally modified form pH2A.X showed a similar sedimentation pattern as the core histones. However, pH2A.X sedimentation in fraction 4 was more pronounced. In Hmga2^{-/-} MEF (Fig. 4a, right and Supplementary Fig. 3b) the levels of pH2A.X were reduced and distributed in fractions 3 to 5. The reducing effect of Hmga2-KO on pH2A.X levels was confirmed by immunostaining in MEF (Supplementary Fig. 4c-d). The subsequent analysis was focused on fractions 3 and 4, because fraction 4 contained all proteins monitored by WB in Hmga2^{+/+} MEF, whereas in fraction 3 levels of HMGA2 and pH2A.X were significantly reduced.”

In addition, we have placed the densitometry analysis of the WB in the new Supplementary Fig. 3b and a longer exposition of the WB in the Western Blot Summary file, which contains all WB presented in the manuscript. Both confirm the description of the results in lines 204-211 mentioned above.

2- Figure 3A and lane 198: the authors conclude to a reduction of pH2A.X levels in Hmga2^{-/-} conditions. How do they come to that conclusion? Looking at the input pH2A.X levels look reduced, but not looking at the fractions: -/- fractions 4 and 5 together could equal fraction 4 in +/+, in which case a redistribution could be another conclusion. Here also a quantification of several blots would be necessary, as well as another method such as immunofluorescence to assess total levels.

We thank Reviewer 2 for this comment. In fact, both interpretations are right. There is a reduction of the pH2A.X levels in Hmga2^{-/-} MEF as demonstrated by the input in the WB (new Fig. 4a), the corresponding densitometry analysis in the (new Supplementary Fig. 3b) and the newly added immunostainings in Hmga2^{+/+} and Hmga2^{-/-} MEF (new Supplementary Figs. 4c-d). In addition, there is a redistribution of the remaining pH2A.X in Hmga2^{-/-} MEF in the fractions 4 and 5. Correspondingly, we have changed the description of this Figure in lines 204-211 in the Results section of the revised manuscript as cited above.

3- Figure 3A and lane 201: related the previous points, the authors decide to focus on fractions 3 and 4 because pH2A.X is present in 4 and absent in 3. We can clearly see pH2A.X in fraction 3 in the ^{-/-} condition, it could therefore be that pH2A.X is distributed differently among fractions based on the different conditions, which could be the reason to focus on these fractions. Once again additional blots, longer exposures and proper quantifications would be necessary.

We agree with this comment from Reviewer 2. We have changed the description of the previous Figure 3a (new Fig. 4a) following the sharp observations from Reviewer 2 as described in our answer to the two previous points. Moreover, we have added new data supporting this new interpretation of the results (new Supplementary Fig. 3b and new Supplementary Figs. 4c-d).

4- Figure 3D and lane 210: the authors conclude of the reduction of SUPT16 and SSRP1 in fraction 4 but not in fraction 3. This is not what the western blot reports: both proteins seems reduced to the same extend in both fractions. As for the other points, additional blots and quantifications would be helpful.

We appreciate this comment from Reviewer 2 that lead us to improve the description of our results. Following the suggestions from Reviewer 2, we have described the results presented in the previous Fig. 3d (new Fig. 4d) in the lines 220 to 223 of the Results section of the revised manuscript as follows:

“WB of SGU fractions using SUPT16- or SSRP1-specific antibodies (Fig. 4d, top) and the corresponding densitometry analysis (Supplementary Fig. 3b) show a reduction of both components of the FACT complex in fractions 3 and 4, being more pronounced in fraction 4.”

Minor comments:

5- Citation of Figure S4F is missing lane 296.

The previous Supplementary Fig.4f is now Supplementary Fig. 6f in the revised version of our manuscript and is cited in line 308. We thank Reviewer 2 for this observation.

Once again, we thank Reviewer 2 for the time and the efforts implemented for the peer review of our manuscript. The comments from Reviewer 2 were mainly focused on protein analysis presented in our manuscript, reflecting her/his expertise and helping us to improve this specific part of our work.

Reviewer #3 (Remarks to the Author):

Transcription and DNA repair were initially considered separate processes occurring at the chromatin. During recent years, however, these processes have been shown to share many protein components, and demonstrated to be intertwined (reviewed e.g. in Fong, Cattoglio and Tijan, 2013, Mol. Cell). Here, Dobersch et al., shed light on the interacting mechanisms of generating DNA nicks, recruiting DNA repair machinery and transcriptional activation of methylated DNA. According to the proposed model, high mobility group protein HMGA2 induces DNA nicks at the TSS of methylated DNA. Subsequently, histone chaperone FACT positions histone variant H2A.X into adjacent nucleosomes, and ATM phosphorylates the H2A.X at the Ser-139. pH2A.X, which hallmarks double-stranded DNA breaks, then attracts DNA-repair machinery to repair the nicks and demethylate the DNA. As a result, the unmethylated DNA can support transcriptional activation of the gene.

The Barreto group has previously shown HMGA2-mediated transcriptional activation to require phosphorylation of H2A.X (Singh et al., 2015; <https://www.nature.com/articles/cr201567>). Moreover, DNA nicks have been linked to demethylation of the DNA and transcriptional activation (reviewed e.g. in Fong, Cattoglio and Tijan, 2013, Mol. Cell). In this study, the authors extend the previous findings by mechanistically dissecting the HMGA2-FACT-ATM-pH2A.X axis at methylated DNA. The study utilizes mouse embryonic fibroblasts (MEFs) derived from wildtype and HMGA2 KO mice, and assays H2A.X positioning, and Pol II levels at the chromatin. Moreover, protein constituents of selected, MNase-digested chromatin fractions are identified using mass spectrometry. To dissect the order of events from HMGA2 binding to demethylation of DNA, the study uses WT and HMGA2 KO mice, inducible HMGA2 mutant with unfunctional AT-hook, chemical inhibition of FACT and ATM, and downregulation of GADD45. The study is in general well conducted and communicated, and the results are up-to date and relevant. However, certain aspects should be clarified before publication.

We appreciate the positive comments from Reviewer 3 towards our work. We thank Reviewer 3 for the time and efforts implemented during the peer review of our manuscript.

To facilitate the recognition of new added results during the revision of our manuscript, we have submitted two pdf files for Reviewers only, in which we highlighted in blue the corresponding text in the manuscript, as well as the figure panels in the main and supplementary figures (please see pdf file containing main text, main and supplementary figures for Reviewers only). Following the constructive suggestions from the Reviewers led us to a substantially improved manuscript.

Major concerns:

1) The general claim of the subtitle “Position of the first nucleosome containing pH2A.X determines the basal transcription” is inadequately backed up by the data. In the study, genes are first ranked based on the pH2A.X levels at the promoter, and the top 15% of genes clustered into three groups using positional information of the pH2A.X (ChIP-seq data). Then, HMGA2 and Pol II with Ser-5 phosphorylated CTD (pPol II) are shown to co-localize with the pH2A.X at these selected genes (Fig. 2E). Finally, transcriptional activity of genes in the three clusters are deduced from mRNA-seq data. Although there is a statistical significance between the mRNA expressions of the distinct groups (Figure 2F), there is a lot of variation within genes, and no causal support for pH2A.X position to determine the transcriptional activity. Please note that the top 15% of genes with the highest pH2A.X level appear in general to be lowly transcribed (based on pPol II ChIP-seq and GRO-seq data, discussed further in points 2 and 3 below). Rather, the data supports a conclusion where pPol II positioning at the TSS correlates with the positioning of the pH2A.X and HMG2A. Moreover, the positions of the nucleosomes (including +1) are not shown. Please, refine the subtitle to match the results and include data (e.g. MNase-seq or H3 ChIP-seq) to report the nucleosomal positions.

We appreciate these very good and constructive comments from Reviewer 3. Following the suggestions from Reviewer 3, we have improved our manuscript in various aspects that we would like to summarize in the following points:

- The subtitle “*Position of the first nucleosome containing pH2A.X determines the basal transcription*” in the previous version of the manuscript was changed to “*Position of the first nucleosome containing pH2A.X correlates with RNA polymerase II and the basal transcription activity of genes*” in the revised version of the manuscript (please see in the Results section in lines 158-159)
- The results related to the ChIP-seq using H3-specific antibodies were incorporated in the revised version of our manuscript (see new Fig. 3a) depicting nucleosome positioning.
- To demonstrate the causal involvement of *Hmga2* in the effects presented in the Figs. 1-3 of the revised version of the manuscript, we have incorporated new results in the Fig 2d, which show that overexpression of *Hmga2* in *Hmga2*^{-/-}MEF reverted the effects caused by *Hmga2*-KO. Please see the description of the new Fig. 2d in lines 167 to 175 in the Results section of the revised version of our manuscript.

2) The quality of the pPol II ChIP-seq data should be adequately demonstrated. Please note that Ser5 phosphorylated Pol II has been shown to extend beyond the promoter-proximal region (e.g. by Kim...Bentley, 2010, cited in this manuscript, and Zaborowska...Murphy, 2016). The spiky pPol II browser shot images (Figure 2C) could be due to the low level of transcription of these example genes (displaying methylated DNA at the promoter). Showing a highly transcribed gene as an example and giving the range of transcriptional activity for the top 15% genes would be helpful in interpreting the results.

These are again constructive observations from Reviewer 3. To address this specific comments from Reviewer 3, we have added the new Supplementary Fig. 2 in the revised version of the manuscript, in which we show reduced basal expression levels (new Supplementary Fig. 2a) and reduced pPol II accumulation (new Supplementary Fig. 2b) in the top 15% candidate genes, when compared to the other genes with higher basal expression levels (top 500).

In addition, visualization of *Hist1h1e* (a gene with higher basal expression level) using the UCSC genome browser showed pPol II accumulation beyond the promoter-

proximal region as previously reported (Kim et al, 2010 and Zaborowska et al, 2016), thereby confirming the quality of our pPol II ChIP-seq data. These new results were mentioned in lines 188 to 190 and the Results section of the revised manuscript.

3) H3 profile is used for detecting +1 nucleosomes in example genes in Figure 1D. Based on these profiles, there seem to be re-positioning of nucleosomes in addition to the reduction in H2A.X levels in HMGA2^{-/-} MEFs. Do the nucleosomal positions show a general change in the absence of HMGA2?

Following Reviewer 3 suggestion, we have included ChIP-seq results using H3-specific antibodies in the revised version of our manuscript (see new Fig. 3a) depicting nucleosome positioning. Based on the results presented in the new Fig. 3a, we cannot conclude that there is a repositioning of the +1 nucleosome. Our results suggest rather a reduction of H3 levels after *Hmga2*-KO that is specific for the top 15% candidate genes, since the H3 ChIP-seq results from the rest of the genes show non-significant reduction of H3 levels (see new Supplementary Figure 2e).

4) The top 15% genes based on pH2A.X levels are shown to display a remarkable loss of pPol II and pH2A.X in the absence of HMGA2 (Figure 2E). Is this loss specific to the top 15% genes or a more general phenomenon across the genome? Please discuss the phenotype of the HMGA2 KO mice and refer to the study generating the mouse.

The reducing effect on pPol II levels after *Hmga2*-KO seems to be specific for the top 15% candidate genes, since the pPol II ChIP-seq results from the rest of the genes show non-significant reduction of pPol II levels (see new Supplementary Figure 2d). The study, in which the *Hmga2*-KO mice was generated is cited in the revised version of the manuscript (see line 132). However, a discussion of the phenotype of this transgenic mice is out of the scope of our manuscript.

5) Lines 317-320 state nascent non-coding RNAs to be found upstream of 46% of the top candidates. Are these ncRNAs divergent transcripts that are generated by

mammalian promoters? See e.g. Jonkers et al., eLife, 2014 that shows 95% of genes in mouse to have divergent transcription. Nevertheless, the nature of these ncRNAs should be disclosed.

This is a very constructive comments from Reviewer 3 that led us to several changes in the revised manuscript, significantly improving the structure of the manuscript and the presentation of our work.

The previous Fig. 4 was divided into Fig. 5 and 6, allowing us to present the DNA/RNA hybrids in the new Fig. 6c with higher magnification and including additional information, such as consensus analysis.

Moreover, to facilitate the understanding of the results presented in the new Figs. 5 and 6, we generated a new Supplementary Fig. 7, which is described in the lines 336-359 as follows:

*“Interestingly, crossing the NONCODE database with our top 15% candidates revealed that 79% of the candidates have annotated noncoding RNAs (ncRNAs) in close proximity ($n = 7,535$), including *Gata6*, *Mtor*, *Igf1* and *Rptor* (Fig. 6c, left top). Mapping the identified ncRNAs to the murine genome allowed us to identify 2,106 unique ncRNAs (7.4%) that mapped to loci close to promoters controlling the expression of adjacent mRNAs (Supplementary Fig. 7b). From these promoter related ncRNAs more than half (1,401; 67%) were in the antisense strand (as) in divergent (div; 621 ncRNAs) or convergent (con; 780 ncRNAs) orientation³⁹ relative to the corresponding promoter and mRNA (Supplementary Fig. 7c-d). Interestingly, *Hmga2*-KO significantly reduced the median expression levels of these antisense divergent ncRNAs from 0.085 to 0.067 ($P = 0.039$; Supplementary Fig. 7e), without significantly affecting the levels of antisense convergent and sense ncRNAs. In silico analysis allowed us to detect putative binding sites of the identified ncRNAs at the TSS of the corresponding mRNAs (Fig. 6c, right) with favorable minimum free energy ($MFE < -55$ kcal/mol) and relatively high consensus (cons > 41%;), supporting the formation of DNA-RNA hybrids containing a nucleotide sequence that favors DNA nicks⁴⁰. In the same genomic regions, we also identified strand asymmetry in the distribution of cytosines and guanines, so called GC skews (Fig. 6c, left middle; Supplementary Fig. 7f-g), that are predisposed to form R-loops, which are three-stranded nucleic acid structures*

consisting of a DNA-RNA hybrid and the associated non-template single-stranded DNA41. Supporting this hypothesis, published genome-wide sequencing experiments after DNA–RNA immunoprecipitation (DRIP-seq) in NIH/3T3 mouse fibroblasts42 confirmed the formation of DNA-RNA hybrids in the top 15% of candidate genes (n = 9,522; Supplementary Fig. 7f-g), including Gata6, Mtor, Igf1 and Rptor (Fig. 6c, left bottom). This correlated with high amounts of GC skews at their TSS with at least 38.5% of the TSS and downstream region having a GC skew higher than 0.05 (n = 3,669).”

6) The section on the R-loops matching the gene’s promoter proximal region requires clarification: Are the ncRNAs aligned to the coding regions of the upstream gene? What is the alignment rate/score? Please note that the DNA-RNA structures (right panels of Fig 4F) are very difficult to see.

Thanks Reviewer 3 for this constructive comments/suggestions. We would like to refer to the answer to the previous point number 5).

Minor comments:

7) The section: “HMGA2 -FACT interaction and -lyase activity are required for pH2A.X deposition and solving of R-loops” is very long. Please consider dividing the HMGA2-FACT interaction and R-loops into their own subtitled sections.

We changed the text in the Results section according to this suggestion from Reviewer 3.

8) Please check the sentence in lines 197-199, it seems to miss something and the logic is hard to follow.

The text in lines 197-199 of the previous version of the manuscript:

“On the other hand, in Hmga2^{-/-} MEF (Figure 3A, right) were only minor variations in the sedimentation pattern of all tested proteins, with the most notable change being the reduction in the levels of HMGA2 and pH2A.X compared to Hmga2^{+/+} MEF.”

Was improved following the suggestion from Reviewer 3 and Reviewer 2 to:

“In Hmga2^{-/-} MEF (Fig. 4a, right and Supplementary Fig. 3b) the levels of pH2A.X were reduced and distributed in fractions 3 to 5. The reducing effect of Hmga2-KO on pH2A.X levels was confirmed by immunostaining in MEF (Supplementary Fig. 4c-d).”

Please see lines 206-209 in the Results section of the revised version of the manuscript.

9) In Figure 3D, DNA is visible only in fractions 3-5. Why isn't DNA detected in the other fractions, such as 1 and 2 that harbor abundant levels of Pol II?

We agree with this observation from Reviewer 3. One should expect DNA in other fractions, such as 1 and 2 that harbor DNA binding proteins. However, a plausible explanation for these results might be that, although there might be DNA in the fractions 1 and 2, the levels of DNA might be relatively low when compared to the DNA levels in fractions 3 to 5.

10) One of the strengths of this study is the disruption of distinct molecules in the HMGA2-FACT-AMT-pH2A.X pathway. However, many inhibitory molecules have side effects and might not be specific to a certain function of the target molecule. Please discuss the mechanism of inhibition and the known selectivity of the FACTin and ATMin. E.g. is it known how CBLC000 trifluoroacetate blocks FACT into a chromatin-bound state and which functions of FACT does the compound inhibit? What is the functional mechanism that KU-55933 exhibits on ATM?

Following this suggestion from Reviewer 3, we have included the mechanism of action of FACTin. Please see lines 271-274 in the revised version of the manuscript.

“FACTin treatment induces negative supercoiling and the formation of left-handed Z-DNA, which is recognized by the FACT subunit SSRP135. Increasing doses of FACTin resulted in chromatin trapping of SUPT16 and SSRP1 (Supplementary Fig. 5c-d).³⁶”

Regarding the mechanism that KU-55933 exhibits on ATM, we have cited the corresponding reference. We did not mention the KU-55933 mechanism in the text because it is not of primary relevance to the concept of the manuscript as the FACTin mechanism is. In addition, the current version of the manuscript is already extensive enough and there is space restrictions... We hope that Reviewer 3 accept our argumentation.

11) Certain sections of the methods should be clarified. E.g. clustering the top 15% genes is vaguely explained (see lines 894-897). Why were only genes with RPKMs less than 10 included in the analyses? This further confuses the claim on pH2A.X determining transcriptional activity (see major concern 1). Likewise, the analyses of downloaded GRO-seq and DRIP-seq data, as well as the R-loop prediction are inadequately described.

Following this suggestion from Reviewer 3 we have improved the corresponding subsections in the Star Methods section of the revised manuscript. Please see the lines 939-943 and 1154-1181.

12) Please consider structuring the discussion with subtitles.

According to this suggestion from Reviewer 3, we have included subtitles in the Discussion section of the revised version of our manuscript. This was another good suggestion from Reviewer 3.

We would like to thank Reviewer 3 once again for the time and the efforts implemented for the peer review of our manuscript. The comments from Reviewer 3 are constructive and helped us to significantly improve the quality of our manuscript.

REVIEWERS' COMMENTS

Reviewer #1 (Remarks to the Author):

My major concerns have been thoroughly addressed in the revised manuscript.

Reviewer #2 (Remarks to the Author):

All my concerns were addressed by the authors.

Reviewer #3 (Remarks to the Author):

The authors have remarkably improved and clarified the manuscript. All the major concerns I had (nucleosomal positioning, expression level of the top 15% genes and the non-coding R-loop RNA) have been sufficiently answered. In specific, the added H3 ChIP-seq data clarifies the questions I had on the nucleosomal positioning. Added analyses on gene expression levels places the results into context, and the nature of ncRNAs are now elaborately discussed. These improvements allow me to support the manuscript for publication.

Sincerely, Anniina Vihervaara, PhD
Academy of Finland Postdoctoral Researcher
University of Helsinki, Finland
Cornell University, Ithaca, NY, USA

From Dec 2020:
Assistant Professor in Gene Technology
Royal Institute of Technology, Stockholm, Sweden
Science for Life Laboratories, Stockholm, Sweden